# Improvement of MPPT Control Performance Using Fuzzy Control and VGPI in the PV System for Micro Grid

**Jong-Chan Kim** [1] **, Jun-Ho Huh** [2] **and Jae-Sub Ko** [3,*]

1   Department of Computer Engineering, Sunchon National University, 255 Jungang-ro, Suncheon-city Jeollanam do 57922, Korea; seaghost@sunchon.ac.kr
2   Department of Data Informatics, Korea Maritime and Ocean University, Busan 49112, Korea; 72networks@kmou.ac.kr
3   Department of Electrical Engineering, Sunchon National University, 255 Jungang-ro, Suncheon-city Jeollanam do 57922, Korea
*   Correspondence: kokos22@sunchon.ac.kr; Tel.: +82-61-750-3540

**Abstract:** This paper proposes the method for maximum power point tracking (MPPT) of the photovoltaic (PV) system. The conventional PI controller controls the system with fixed gains. Conventional PI controllers with fixed gains cannot satisfy both transient and steady-state. Therefore, to overcome the shortcomings of conventional PI controllers, this paper presents the variable gain proportional integral (VGPI) controllers that control the gain value of PI controllers using fuzzy control. Inputs of fuzzy control used in the VGPI controller are the slope from the voltage-power characteristics of the PV module. This paper designs fuzzy control's membership functions and rule bases using the characteristics that the slope decreases in size, as it approaches the maximum power point and increases as it gets farther. In addition, the gain of the PI controller is adjusted to increase in transient-state and decrease in steady-state in order to improve the error in steady-state and the tracking speed of maximum power point of the PV system. The performance of the VGPI controller has experimented in cases where the solar radiation is constant and the solar radiation varies, to compare with the performance of the P&O method, which is traditionally used most often in MPPT, and the performance of the PI controller, which is used most commonly in the industry field. Finally, the results from the experiment are presented and the results are analyzed.

**Keywords:** PV system; PI controller; fuzzy control; MPPT; tracking speed; error; Micro Grid

## 1. Introduction

Renewable energy is drawing much attention as an energy source that can replace fossil fuels. Solar energy is the most representative renewable energy and an infinite, eco-friendly energy but is highly dependent on temperature and solar radiation. Temperature affects voltage, and solar radiation affects current [1–3]. Temperature and solar radiation have a direct impact on solar power output and, in particular, cause a change in MPP (maximum power point). In order to track the MPP of the PV system efficiently, an appropriate DC-DC converter and a tracking algorithm (method) must be integrated and configured, and the following conditions must be satisfied [4]:

- Fast tracking response (transient response).
- No vibration around the MPP (steady-state response).
- Response performance against insolation and temperature change.
- Simple structure and low cost.

Typical methods for MPPT (maximum power point tracking) are the Constant Voltage method using a fixed ratio of the open voltage, P&O (Perturb & Observe) method using power and voltage perturbation, and IncCond (Incremental Conductance) method using slope and conductance that can be obtained in the current ($I_{pv}$)-voltage ($V_{pv}$) and power (P)-voltage ($V_{pv}$) curves [5–13]. In addition, These methods use reference voltage [14–21], reference current [22,23], or duty ratio [24] for maximum power point tracking. Among them, the P&O method has the advantages of simple structure and low calculation, whereas the IncCond method has the advantage of tracking the MPP faster and more accurately than the P&O. These methods are most generally used for the MPPT of the photovoltaic (PV) system due to the aforesaid advantages. Since the P&O and IncCond methods track the MPP while varying the voltage or current by a fixed size, however, vibration may occur near the MPP, and performance may deteriorate; thus rapidly changing the solar radiation conditions. Although used to solve these problems, artificial neural networks also have problems such as long learning time and high computational complexity [25,26].

Therefore, this paper proposes a method of tracking the MPP of the PV system using the PI controller, which is widely used to control the industrial field [27–33]. The PI controller is a controller that uses proportional gain and integral gain. The proportional gain and the integral gain of the PI controller are closely related to the rise time, settling time, and steady-state error, and there is clear relationship between the gain and the control amount. In addition, it has a simple structure and a small amount of calculation, which enables a quick response. Since the PI controller is generally controlled with a fixed gain value, however, it is difficult to satisfy both transient and steady states. The response performance of the steady state is degraded when the gain value is increased for the fast response of the transient state, whereas, the performance of the transient state is degraded if the gain value is reduced to improve the steady-state response performance. Therefore, it is necessary to control the gain so as to satisfy both transient and steady states by automatically adjusting the gain value of the PI controller according to the operation state. In this paper, fuzzy control is used to adjust the gain of the PI controller. Methods for adjusting the gain value of PI controller using fuzzy control were presented through several studies [34–38]. However, existing studies depend on designer knowledge for rule base and membership function designs and do not suggest how the gain value of a PI controller changes with its operational state. Thus, the paper proposes simple and clear fuzzy control membership function and rule base design according to the characteristics of the PV system and shows the gains of PI controller, which are changing in the transient and steady-state of the system.

Fuzzy control does not require accurate modeling and has the advantage of controlling nonlinear systems [39]. The VGPI (variable gain proportional integral) controller proposed in this paper uses the voltage and current of photovoltaic power generation, as inputs to control the gain value of the PI controller with a fuzzy controller, and the PI controller tracks the MPP of the PV system.

The VGPI controller compares the error at tracking speed and steady state with the most commonly used P&O controller and PI controller with a fixed gain value. It also shows the characteristics of gain values of the PI controller controlled by fuzzy control in the VGPI controller.

The rest of this paper is organized as follows: Section 2 discusses the DC-DC converter and the conventional MPPT method; Section 3 presents the MPPT control by the VGPI controller; Section 4 shows the comparison and analysis results of the MPPT control characteristics with the method proposed in the paper and the existing method; Finally, Section 5 presents the conclusion.

## 2. Conventional MPPT Method

### 2.1. MPPT Control by DC-DC Converter

Figure 1 shows the MPPT control of photovoltaic power generation using the DC-DC converter [40–43]. The PV module supplies voltage and current to load R through the DC-DC converter. Figure 2 shows the relationship curves of the current ($I_{pv}$) - voltage ($V_{pv}$) of the PV module.

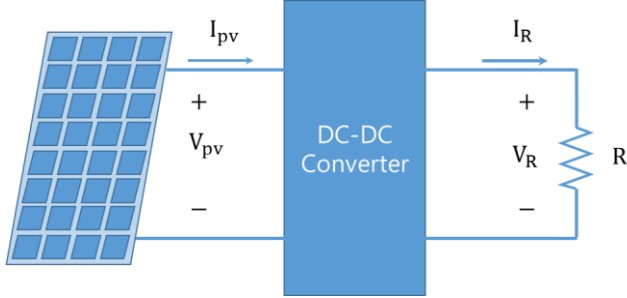

**Figure 1.** Maximum power point tracking (MPPT) control of the photovoltaic (PV) system by DC-DC converter.

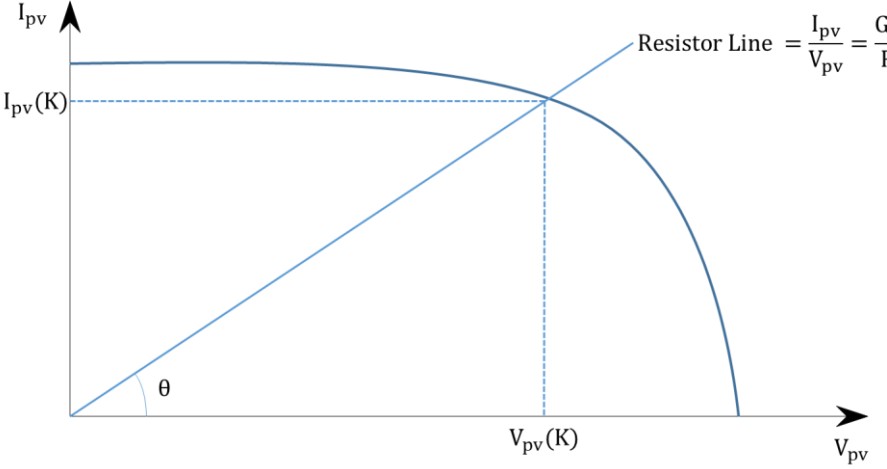

**Figure 2.** I-V curve of the PV module.

If the gain of the converter is G, the relationship between input and output in Figure 2 can be obtained as follows:

$$V_R = R \times I_R \tag{1}$$

$$G = \frac{V_R}{V_{pv}} \tag{2}$$

$$G = \frac{I_{pv}}{I_R} \tag{3}$$

$$\frac{V_{pv}}{I_{pv}} = \frac{R}{G^2} \tag{4}$$

$$V_{pv} = \frac{R}{G^2} \times I_{pv} \tag{5}$$

In Figure 2, the inclination angle ($\theta$) of the resistor line can be calculated as follows:

$$\theta = \text{atan}\left(\frac{G^2}{R}\right) \tag{6}$$

In this paper, a buck converter is used as DC-DC converter. Figure 3 shows the structure of the buck converter.

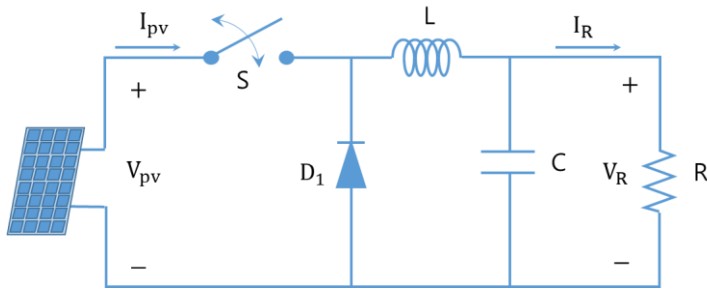

**Figure 3.** Structure of the buck converter.

The diode ($D_1$) is reverse-biased during on-time and forward-biased during off-time in the buck converter. The relationship between the input and output of the Buck converter is expressed by the duty ratio (D) [44–46].

$$\frac{V_R}{V_{pv}} = D \tag{7}$$

$$\frac{I_R}{I_{pv}} = \frac{1}{D} \tag{8}$$

Since the buck converter has the same gain (G) and duty ratio, it can be expressed as:

$$D = G \tag{9}$$

$$\theta = \text{atan}\left(\frac{G^2}{R}\right) = \text{atan}\left(\frac{D^2}{R}\right) \tag{10}$$

Since the duty ratio of the buck converter is $0 \leq D \leq 1$, the range of the inclination angle can be determined as follows, and Table 1 shows the maximum and minimum inclination angles of the buck converter:

$$\theta|_{D=0} = \text{atan}\left(\frac{0^2}{R}\right) = 0 \tag{11}$$

$$\theta|_{D=1} = \text{atan}\left(\frac{1^2}{R}\right) = \text{atan}\left(\frac{1}{R}\right) \tag{12}$$

**Table 1.** Maximum and minimum inclination angles.

| Minimum Inclination Angle | Maximum Inclination Angle |
|:---:|:---:|
| $\theta|_{D=0} = 0$ | $\theta|_{D=1} = \text{atan}\left(\frac{1}{R}\right)$ |

Figure 4 shows the tracking region and non-tracking region of the MPP according to the inclination angle. In order to track the MPP efficiently, the resistance value must be selected such that the inclination angle is lower than that at the MPP. When the duty ratio of the buck converter changes from 0 to 1, the buck converter changes from the maximum voltage ($V_{oc}$), the open voltage, to the load voltage ($V_R$) [4].

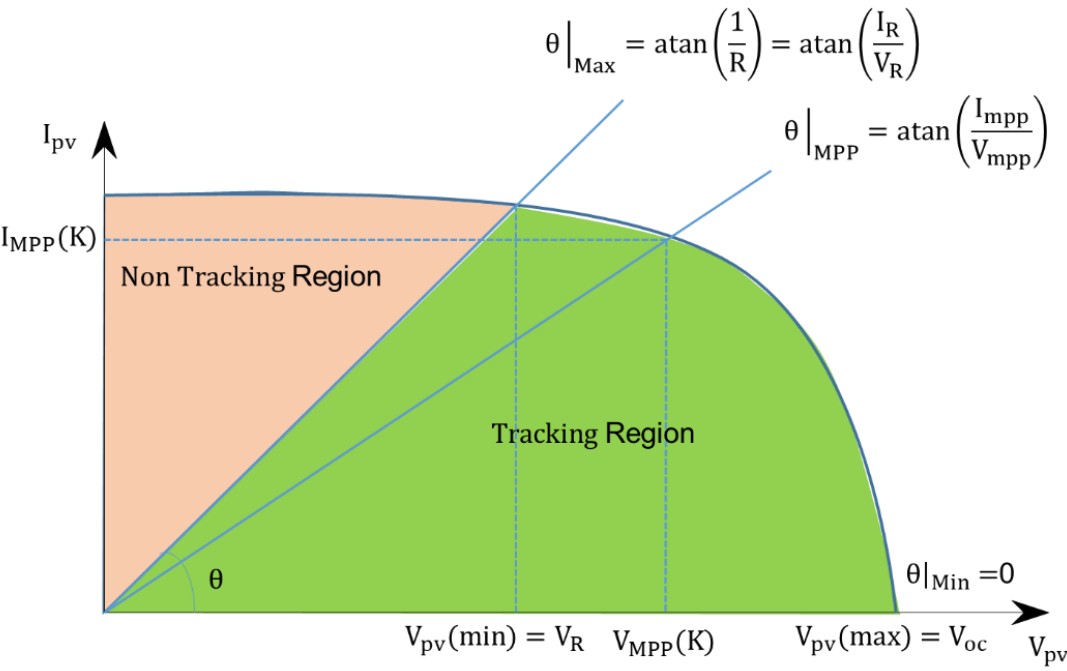

$$\theta\Big|_{Max} = atan\left(\frac{1}{R}\right) = atan\left(\frac{I_R}{V_R}\right)$$

$$\theta\Big|_{MPP} = atan\left(\frac{I_{mpp}}{V_{mpp}}\right)$$

**Figure 4.** MPPT range according to the inclination angle.

### 2.2. P&O Method

The P&O MPPT method tracks the MPP by varying the voltage of the solar cell and observing the power and increasing or decreasing the PV voltage in the direction wherein the current power is greater than the previous power.

Table 2 shows the operating state of the P&O method according to the voltage and power states. In cases 1 and 3, when voltage change (ΔV) is increasing (positive) or decreasing (negative), power change (ΔP) is increasing (positive), and control is continued in the same direction. In cases 2 and 4, since power change (ΔP) is negative, it tracks the maximum power by performing control in the direction opposite to the change in voltage. Figure 5 shows the flow chart of Table 2 [5,6,10,15].

**Table 2.** Operating state of the P&O method according to the voltage and power states.

| Case | Perturbation $[\Delta V_{pv}=V_{pv}(k)-V_{pv}(k-1)]$ | Change in Power $[\Delta P=P(k)-P(k-1)]$ | Next Perturbation $[\Delta V_{ref}(k)]$ |
|------|------|------|------|
| 1 | Positive | Positive | Positive |
| 2 | Positive | Negative | Negative |
| 3 | Negative | Positive | Negative |
| 4 | Negative | Negative | Positive |

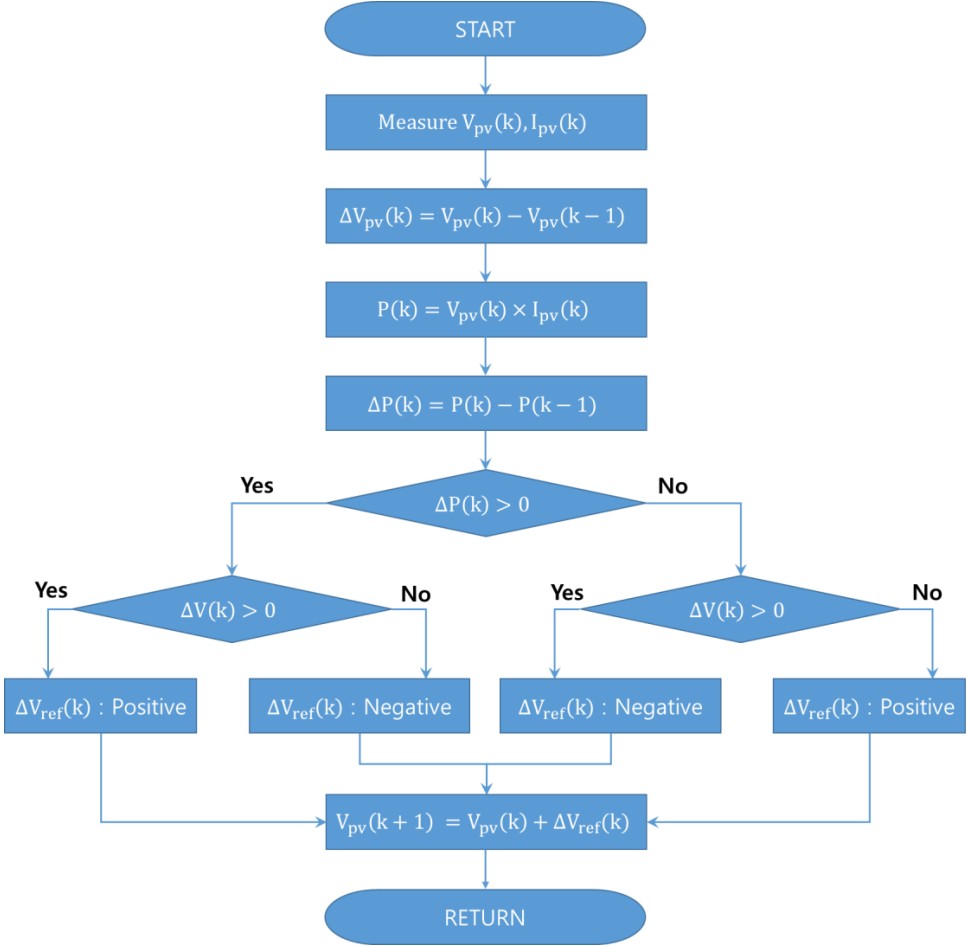

**Figure 5.** Flowchart of the Perturb & Observe (P&O) MPPT method.

### 2.3. IncCond Method

The IncCond (Incremental Conductance) method is a method of using the slope of the power-voltage curve of the solar cell, and the slope can be expressed by Equation (20). It is one of the most widely used methods in the field along with the P&O method because of its stable characteristics and simple implementation method. Figure 6 shows the control principle of IC MPPT. As shown in the characteristic curve of Figure 6, IC MPPT finds the MPP by using the fact that the slope of the characteristic curve is 0 (zero) in MPP. The slope of the output curve of the solar cell can be expressed as $dP/dV_{pv}$ . In Figure 6, the MPP is point B, and the slope is zero. Based on the MPP, we can see that the left side has a positive slope, and the right side has a negative slope. The conditions at each point are shown in Equations (14)–(16). In Equations (14)–(16), $I_{pv}/V_{pv}$ is the conductance of the inverse of the resistance, and $dI_{pv}/dV_{pv}$ is the incremental conductance of the change in conductance. Therefore, the method of using the slope of power and voltage is called the incremental conductance (IncCond) method.

$$\frac{dP}{dV_{pv}} = \frac{d\left(V_{pv}I_{pv}\right)}{dV_{pv}} = \frac{dV_{pv} \times I_{pv}}{dV_{pv}} + \frac{V_{pv} \times dI_{pv}}{dV_{pv}} = I_{pv} + V_{Pv}\frac{dI_{pv}}{dV_{pv}} \tag{13}$$

$$A : \frac{dP}{dV_{pv}} = I_{Pv} + V_{pv}\frac{dI_{pv}}{dV_{pv}} < 0 \rightarrow \frac{dI_{pv}}{dV_{pv}} < -\frac{I_{pv}}{V_{pv}} \tag{14}$$

$$B : \frac{dP}{dV_{pv}} = I_{Pv} + V_{pv}\frac{dI_{pv}}{dV_{pv}} = 0 \rightarrow \frac{dI_{pv}}{dV_{pv}} = -\frac{I_{pv}}{V_{pv}} \tag{15}$$

$$C: \frac{dP}{dV_{pv}} = I_{Pv} + V_{pv}\frac{dI_{pv}}{dV_{pv}} > 0 \rightarrow \frac{dI_{pv}}{dV_{pv}} > -\frac{I_{pv}}{V_{pv}} \tag{16}$$

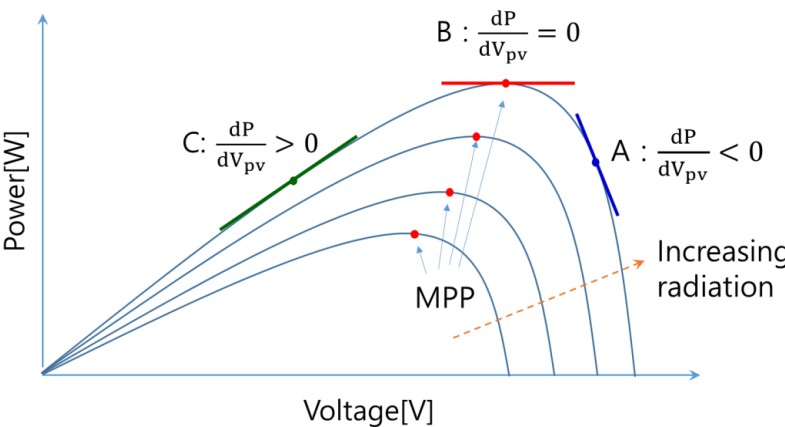

**Figure 6.** Control principle of the IncCond MPPT method.

In Figure 6, the MPP of the photovoltaic power generation moves to the left as the solar radiation increases [47,48]. Therefore, if the solar radiation is increased, the voltage is increased; if the solar radiation is decreased, however, the voltage is decreased, and the MPP change due to the solar radiation change can be tracked more quickly.

The flow chart of the IncCond method is shown in Figure 7 using the slope condition of the P-V curve and MPP variation according to the changing solar radiation in Figure 6 [12,14].

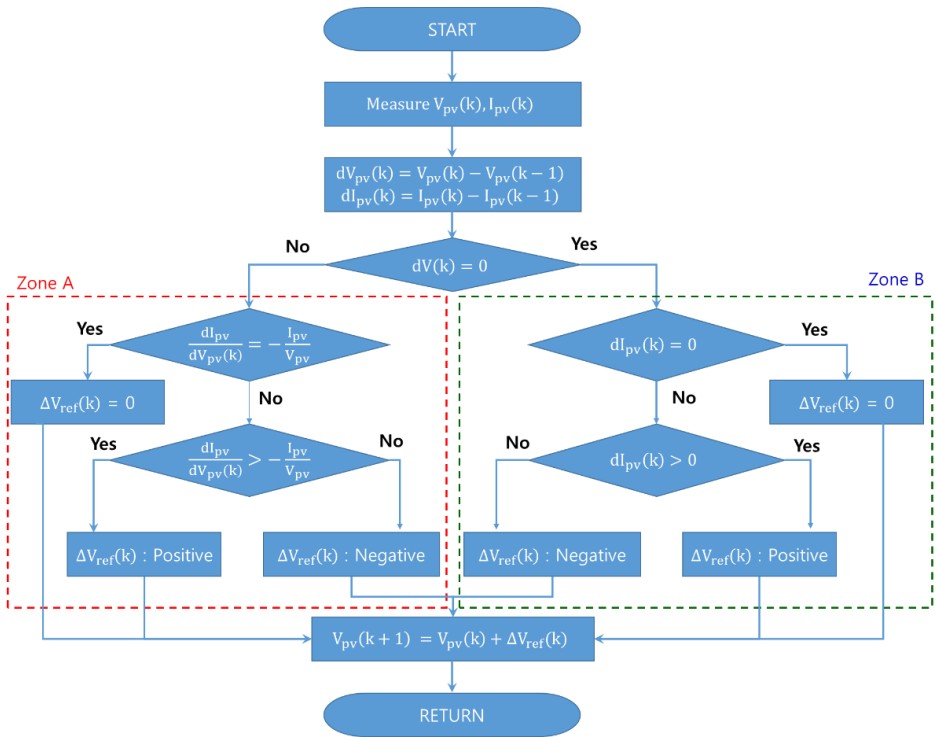

**Figure 7.** Flow chart of the IncCond method.

In Figure 7, Zone A shows the part that tracks the MPP along the slope in the P-V curve, and Zone B shows the part that tracks the MPP for the change in solar radiation. Since solar radiation greatly affects the current of the photovoltaic power generation, the change of solar radiation is caused by the

change of current. Therefore, the amount of solar radiation can be judged to have changed when only the change of current occurs without a change in voltage.

## 3. Proposed MPPT Method

In this paper, a PI controller is used to track the MPP of the PV system. The PI controller is a method that uses proportional control and integral control. The PI controller uses the gain for proportional control and the gain for integral control. Table 3 shows the effect of these gain values on the system. If large values of proportional gain and integral gain are selected to reduce the rise time and steady-state error, the overshoot may increase greatly, and the settling time may be longer. In addition, the error may increase in steady state. Generally, in the case of PI control, since two gain values are fixed, it is very important to select the gains value corresponding to the control state [27,49].

**Table 3.** Influence of the proportional integral (PI) controller gain value on the system.

| Parameter | Rise Time | Overshoot | Settling Time | Steady-State Error |
|-----------|-----------|-----------|---------------|--------------------|
| $K_p$ Increase | Decrease | Increase | Small Change | Decrease |
| $K_i$ Increase | Decrease | Increase | Increase | Decrease Significantly |

This paper proposes a method of adjusting the gain value of the PI controller using fuzzy control to improve this characteristic of the PI controller. Fuzzy control does not require accurate system modeling and has the advantage of handling nonlinear systems. Fuzzy control is controlled by using the rules of the "IF THEN" structure in simple language. The fuzzy controller inputs the error and the error change value to perform control through fuzzification and inferential engine defuzzification [28]. The most common reasoning method used in the fuzzy controller is Mamdani's MIN-MAX method. The "IF THEN" rule for multiple inputs has "AND" and "OR" operations, and it can be expressed as follows [4]:

$$\text{IF x is } A_1 \text{ AND x is } A_2 \ldots \text{ AND x is } A_n \text{ THEN y is } B_s$$
$$\text{IF x is } A_s \text{ THEN y is } B_s$$
$$A_s = A_1 \cap A_2 \cap A_3 \cap \ldots A_n \tag{17}$$
$$\mu A_s(x) = \text{Min}[\mu A_1(x), \ \mu A_2(x), \cdots, \mu A_n(x)]$$

$$\text{IF x is } A_1 \text{ OR x is } A_2 \ldots \text{ OR x is } A_n \text{ THEN y is } B_s$$
$$\text{IF x is } A_s \text{ THEN y is } B_s$$
$$A_s = A_1 \cup A_2 \cup A_3 \cup \ldots A_n \tag{18}$$
$$\mu A_s(x) = \text{max}[\mu A_1(x), \ \mu A_2(x), \cdots, \mu A_n(x)]$$

where $\mu A_n(x)$ represents the membership strength of the fuzzy membership function for input $A_n$. Various methods of adjusting the gain value of the PI controller using fuzzy control have been proposed. These methods are based on user knowledge in designing Fuzzy Control's membership functions and rule base and do not represent the background to design. This approach has the problem of redesigning membership functions and rule bases for other users to use.

Therefore, in this paper, using the operating characteristics for the gain value of the PI controller, a simpler and easier-to-understand controller is designed. Member functions and rule bases designed in the paper are based on the following:

1.  Error and changing error which is the input of fuzzy control, are as shown in expressions (19) and (20).
2.  The error and the changing error decrease in size as solar power is closer to the MPP.
3.  If the input value is large, the tracking speed should be fast because it is far from the MPP. This increases the gain value of the PI controller.

4.  If the input value is small, it is close to the MPP and the error in steady state must be reduced.

This reduces the gain value of the PI controller.

The input of fuzzy control, error and error variation, are divided into seven sections: Negative big (NB), Negative medium (NM), Negative small (NS), zero(ZE), Positive big (PB), Positive medium (PM) and positive small (PS). The output of the fuzzy control is designed to perform three actions: increase (P: positive), hold (ZE: zero) and decrease (N: negative).

Tables 4 and 5 show the rule base for proportional gain ($K_p$) and integral gain ($K_i$) designed in the paper, and Figures 8–10 show membership function for the input and output of fuzzy control.

**Table 4.** Rule base to adjust gain $K_p$.

| ce \ e | NB | NM | NS | ZE | PS | PM | PB |
|---|---|---|---|---|---|---|---|
| NB | P | P | ZE | ZE | ZE | P | P |
| NM | P | ZE | ZE | N | ZE | ZE | P |
| NS | P | P | N | N | N | P | P |
| ZE | P | ZE | N | N | N | ZE | P |
| PS | P | P | N | N | N | P | P |
| PM | P | ZE | ZE | N | ZE | ZE | P |
| PB | P | P | ZE | ZE | ZE | P | P |

**Table 5.** Rule base to adjust gain $K_i$.

| ce \ e | NB | NM | NS | ZE | PS | PM | PB |
|---|---|---|---|---|---|---|---|
| NB | P | P | ZE | ZE | ZE | P | P |
| NM | P | P | ZE | N | ZE | P | P |
| NS | P | P | N | N | N | P | P |
| ZE | P | ZE | N | N | N | ZE | P |
| PS | P | P | N | N | N | P | P |
| PM | P | P | ZE | N | ZE | P | P |
| PB | P | P | ZE | ZE | ZE | P | P |

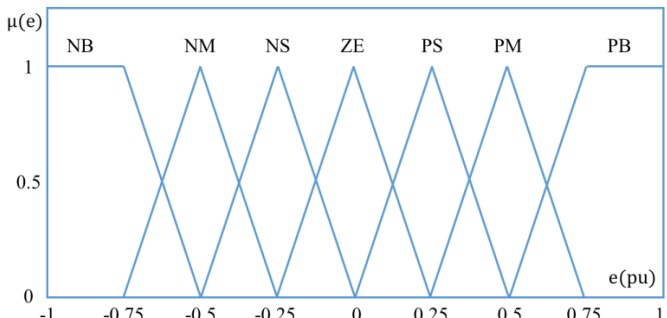

**Figure 8.** Member function for error.

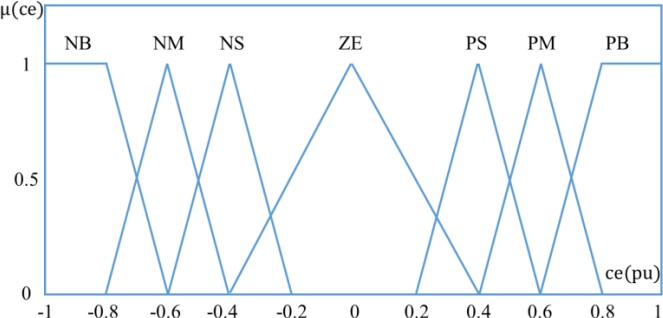

**Figure 9.** Member function for changing error (ce).

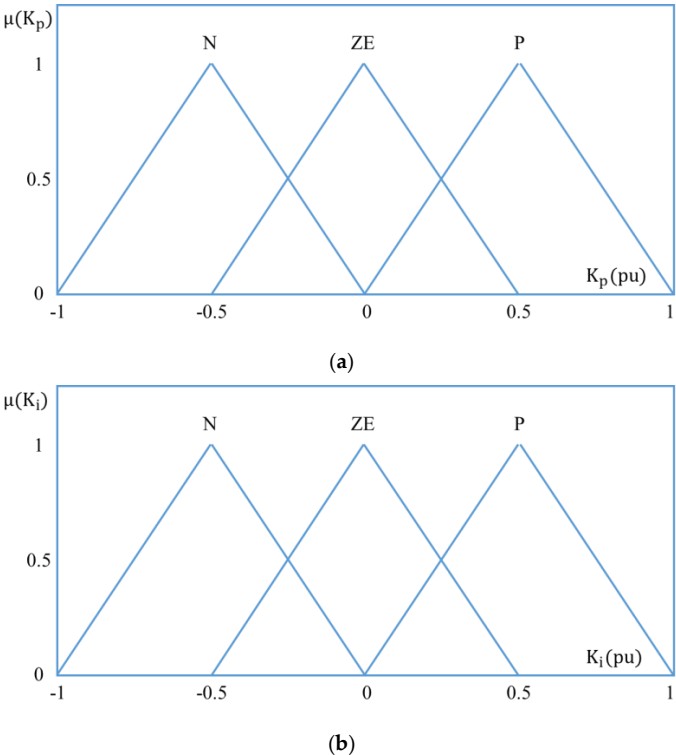

**Figure 10.** Member function for output. (**a**) Proportional gain ($K_p$); (**b**) Integral gain ($K_i$).

$$E(k) = \frac{P(k) - P(k-1)}{V_{pv}(k) - V_{pv}(k-1)} \tag{19}$$

$$CE(k) = E(k) - E(k-1) \tag{20}$$

Equations (21)–(24) show the gain of the PI controller adjusted by the fuzzy controller. Outputs $\Delta K_p$ and $\Delta K_i$ of the fuzzy controller are calculated using the center of gravity (COG) [27,50]:

$$K_p(k) = K_p(k-1) + \Delta K_p \tag{21}$$

$$K_i(k) = K_i(k-1) + \Delta K_i \tag{22}$$

$$\Delta K_p = \frac{\sum_{j=1}^{n} \mu\left(K_p\right)_j \cdot \left(K_p\right)_j}{\sum_{j=1}^{n} \mu\left(K_p\right)_j} \tag{23}$$

$$\Delta K_i = \frac{\sum_{j=1}^{n} \mu(K_i)_j \cdot (K_i)_j}{\sum_{j=1}^{n} \mu(K_i)_j} \tag{24}$$

Figure 11 shows an example of the input error (0.7) and changing error (0.3) of fuzzy control. When the error and error change values are calculated, the membership strength is calculated from the membership functions shown in Figures 8 and 9. In the membership function for error 0.7, the membership strength is 0.8 for PM and 0.2 for PB. For changing error 0.3, the strength of membership is calculated for the membership function, which is 0.5 for ZE and 0.25 for PS. Four output values are calculated through the AND operation of Equation (17) and the rule base of $K_p$ in Table 4, and 0.413 can be obtained by calculating the final values through Equation (23).

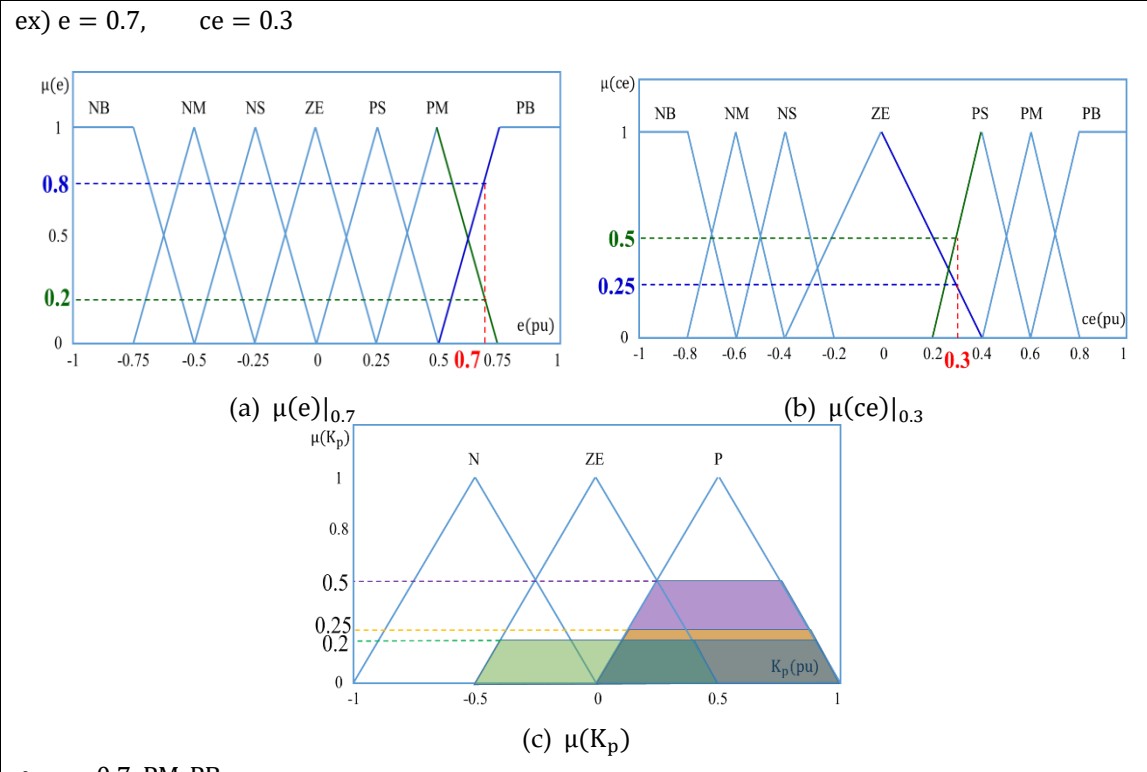

(a) $\mu(e)|_{0.7}$ 　　　(b) $\mu(ce)|_{0.3}$

(c) $\mu(K_p)$

- $e = 0.7$: PM, PB
- $ce = 0.3$: ZE, PS
- $\mu e_1(0.7) = 0.2$ (PM)
- $\mu e_2(0.7) = 0.8$ (PB)
- $\mu ce_1(0.3) = 0.25$ (ZE)
- $\mu ce_2(0.3) = 0.5$ (PS)
- $j = 1$  IF e is $\mu e_1(0.7)$: PM AND ce is $\mu ce_1(0.3)$: ZE THEN $K_p$ is ZE

  $\mu(K_p)_1 = \text{Min}(\mu e_1(0.7), \mu ce_1(0.3)) = \text{Min}(0.2, 0.25) = 0.2$

  $K_{p_1} = \text{ZE}$
- $j = 2$  IF e is $\mu e_1(0.7)$: PM AND ce is $\mu ce_2(0.3)$: PS THEN $K_p$ is P

  $\mu(K_p)_2 = \text{Min}(\mu e_1(0.7), \mu ce_2(0.3)) = \text{Min}(0.2, 0.5) = 0.2$

  $K_{p_2} = \text{P}$
- $j = 3$  IF e is $\mu e_2(0.7)$: PB AND ce is $\mu ce_1(0.3)$: ZE THEN $K_p$ is P

  $\mu(K_p)_3 = \text{Min}(\mu e_2(0.7), \mu ce_1(0.3)) = \text{Min}(0.8, 0.25) = 0.25$

  $K_{p_3} = \text{P}$
- $j = 4$  IF e is $\mu e\_2 (0.7)$ : PB AND ce is $\mu ce\_2 (0.3)$ : PS THEN $K\_p$ is P

  $\mu(K_p)_4 = \text{Min}(\mu e_2(0.7), \mu ce_2(0.3)) = \text{Min}(0.8, 0.2) = 0.5$

  $K_{p_3} = \text{P}$
- Set the representative value of $K_p$ as follows:

|  | N | ZE | P |
|---|---|---|---|
| **Representative value** | -0.5 | 0 | +0.5 |

$$\Delta K_p = \frac{\sum_{j=1}^{4} \mu(K_p)_j \cdot (K_p)_j}{\sum_{j=1}^{4} \mu(K_p)_j} = \frac{0.2 \times 0 + 0.2 \times 0.5 + 0.25 \times 0.5 + 0.5 \times 0.5}{0.2 + 0.2 + 0.25 + 0.5} = 0.413$$

**Figure 11.** $K_p$ calculation for error and changing error.

Figure 12 shows the structure of the VGPI controller for MPPT control of the PV system. The inputs are the voltage ($V_{pv}$) and current ($I_{pv}$) of the PV system, and the output is the change value of the PI controller gain value ($\Delta K_p, \Delta K_i$) through the fuzzy controller. The PI controller outputs the PWM (Pulse Width Modulation) signal for MPPT control using the adjusted gains by fuzzy control, and this signal controls the DC-DC converter.

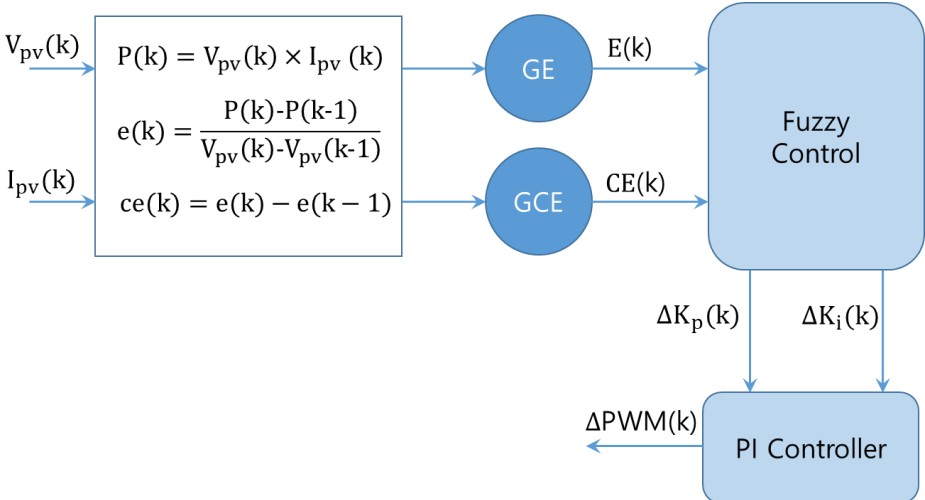

**Figure 12.** Variable gain proportional integral (VGPI) controller for the MPPT control of the PV system.

## 4. Experiment Result

The control performance of MPPT is verified by the speed at which maximum power is tracked and the magnitude of the error in steady state using voltage, current and power output from the PV module. In order to verify MPPT's performance in the paper, the experimental device was constructed with a PV module, a Buck converter, a DC-DC step down converter and a battery.

Experiments in solar power use solar simulators or use artificial light sources to construct a constant experimental environment. In the paper, a constant experimental environment was constructed using artificial lighting. Artificial light sources in the experimental environment can be used to maintain or change the test conditions. In addition, the same environmental conditions can be configured for different methods, so that the performance of the proposed method and the conventional method can be compared. The proposed method and conventional method compare the speed at which the maximum power point is tracked and the error at steady state. Since the environment is constructed using the same artificial light source, comparisons of output power, voltage and current can be a valid method for verifying peak power point tracking performance [51–54].

Figure 13 shows the circuit diagram and control system for the MPPT control performance test of solar power generation. In this paper, MPPT control is controlled by the buck converter, and voltage and current are measured using the INA219 voltage current sensor. Switching of the buck converter was performed using P-channel MOSFET (F9530N) for high-side switching of the buck converter. P-channel MOSFET has a switching state of "on" when a "low" signal is inputted to the gate, so the NPN transistor (2N3904) and pull-up resistor (1 kΩ) are used to control the buck converter.

DC-DC step down converter (KIS-3R33S) was used to maintain constant voltage for changes in the voltage of solar power, and the cell phone battery was charged.

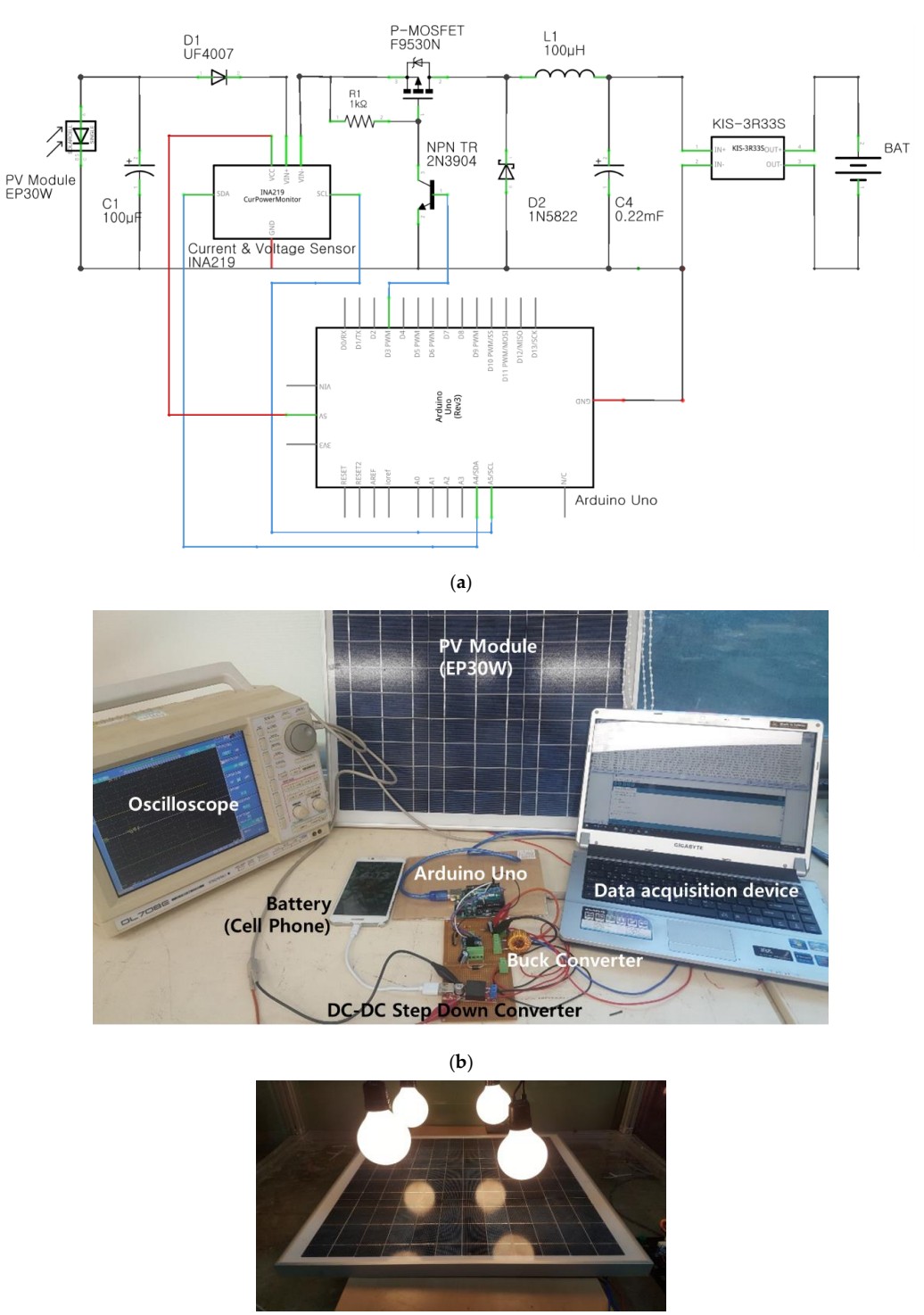

(a)

(b)

(c)

**Figure 13.** Experimental setup for the MPPT control performance test of the PV system. (**a**) Circuit diagram for experiments; (**b**) System for experiments; (**c**) Artificial light.

Figure 14 shows the change of output voltage according to PWM signal of Buck Converter. When MPPT control is performed using a buck converter, the voltage gradually decreases from the open-circuit voltage to the load voltage according to the PWM signal. In Figure 14, CH1 represents the PWM signal output from the controller, and CH2 denotes the voltage change of the PV module. The switching frequency of the controller used is 3.9 [kHz].

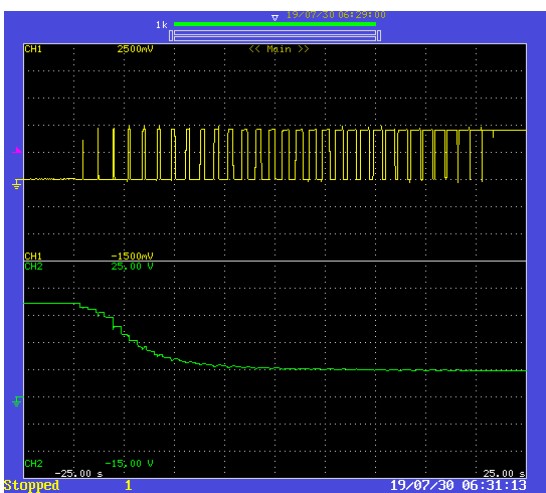

**Figure 14.** PWM signal and voltage PV module ($V_{pv}$).

Figures 15–17 show the response characteristics of the VGPI, PI, and P&O methods under constant solar irradiation conditions.

The PI controller used for comparison with VGPI uses 0.035 for proportional gain ($K_p$) and 0.005 for integral gain ($K_i$). Arduino's PWM ranges from 0 to 255, with 0 representing 0% and 255 representing 100% duty ratio. The P&O controller adjusts the PWM to a fixed size 3 to regulate the voltage at a constant rate.

In Figure 15, Figure 15a shows the voltage and current, Figure 15b presents the output power, Figure 15c illustrates the gain of the PI controller controlled by fuzzy control, Figure 15d shows the control value ($C_p$) and PWM signal for switching control of the DC-DC converter and Figure 15e is output voltage controlled by step down converter. The gain of the PI controller in (C) is increased for fast tracking in transient state, and the gain value is decreased for improving accuracy and stability in steady state. The control value ($C_p$) for tracking the MPP increases as the gain of the PI controller is adjusted according to the operating state and decreases in steady state. As a result, the variation of the PWM signal for switching of the DC-DC converter is reduced, and the power ripple is reduced; thus enabling more accurate MPPT. The output voltage in Figure 15e remains constant even as the voltage of solar power changes.

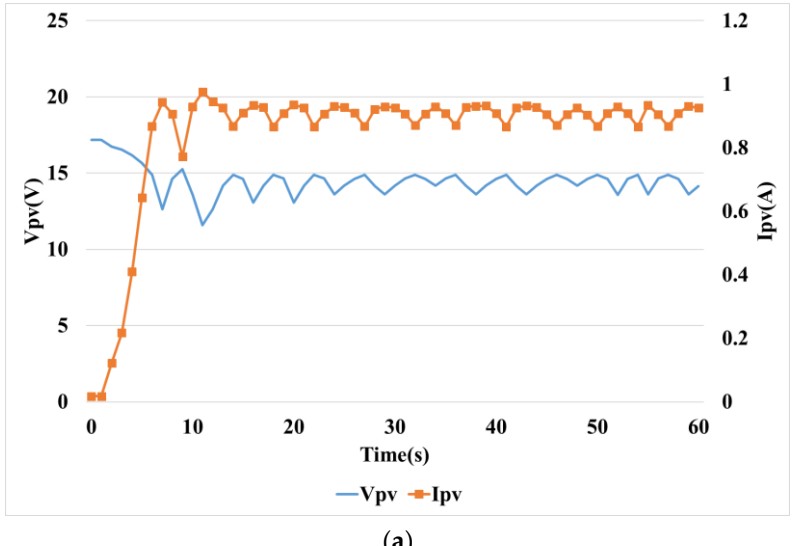

(**a**)

**Figure 15.** *Cont.*

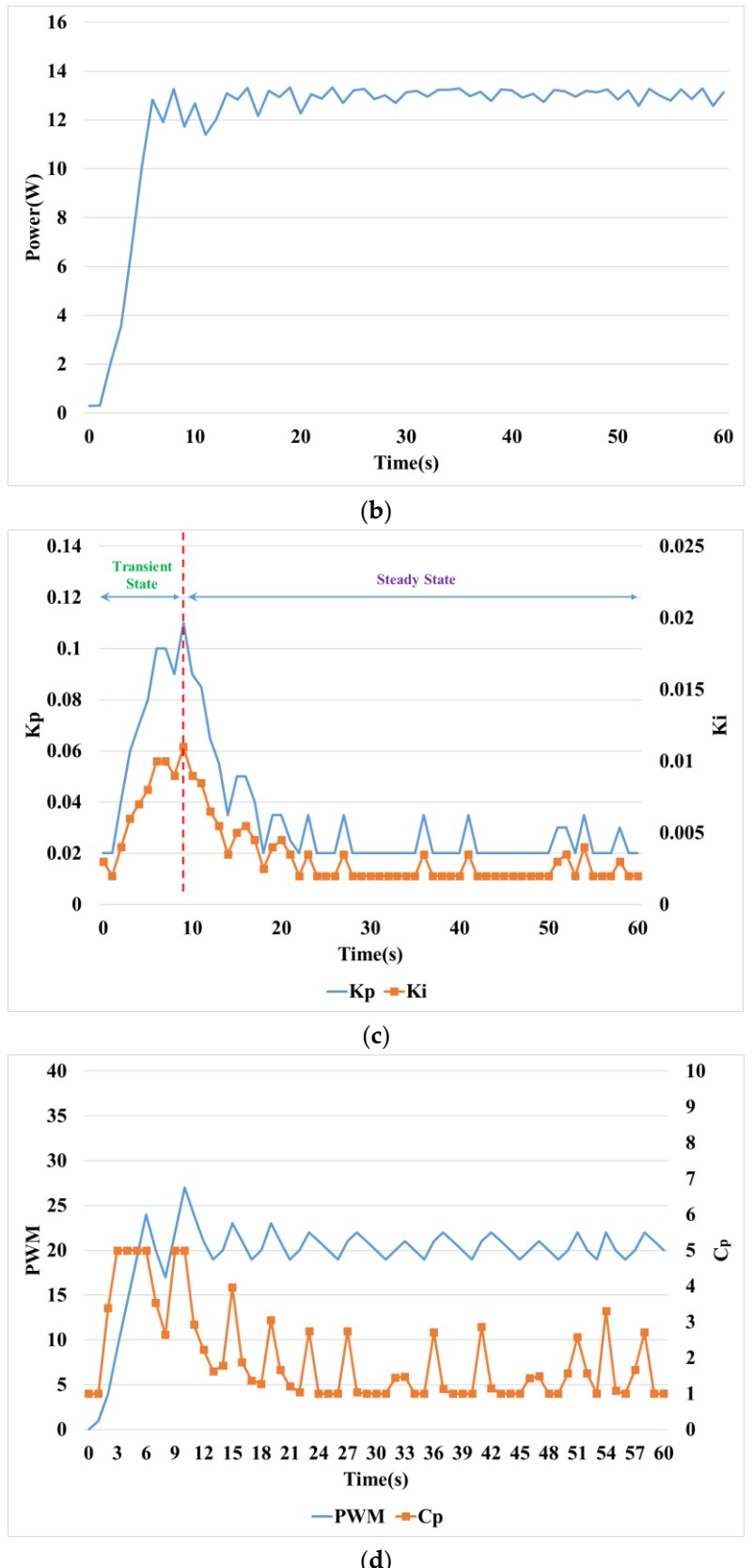

**Figure 15.** *Cont.*

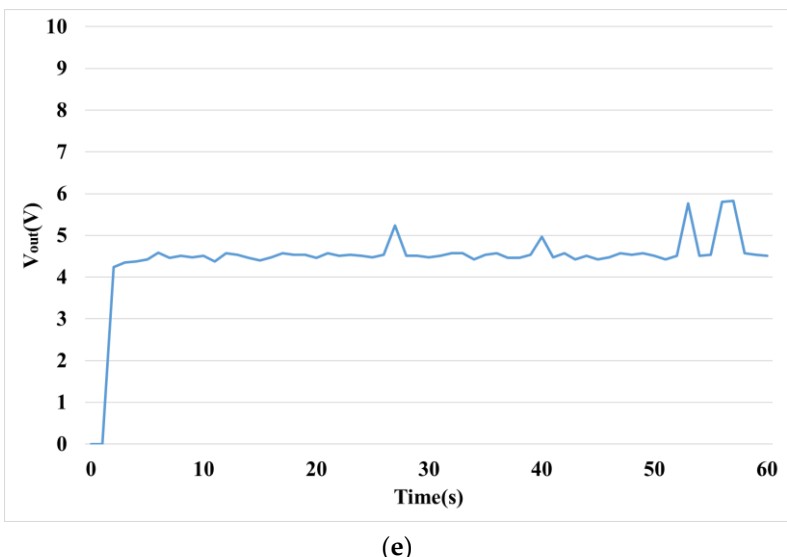

(**e**)

**Figure 15.** Response characteristics of the VGPI MPPT method. (**a**) Voltage ($V_{pv}$) and Current ($I_{pv}$) of the PV module; (**b**) Output Power of the PV module; (**c**) Proportional gain ($K_p$) and Integral gain ($K_i$) of the PI controller; (**d**) PWM signal and control value ($C_p$) (**e**) Output Voltage.

Figure 16 shows the response performance of the MPPT control of photovoltaic power generation using the PI controller. In particular, Figure 16c shows the fixed gain of the PI controller. Although the PWM signal and the control value ($C_p$) of Figure 16d are controlled according to operating state by PI control, the ripple of the output power increases because it is larger than the value of Figure 15d.

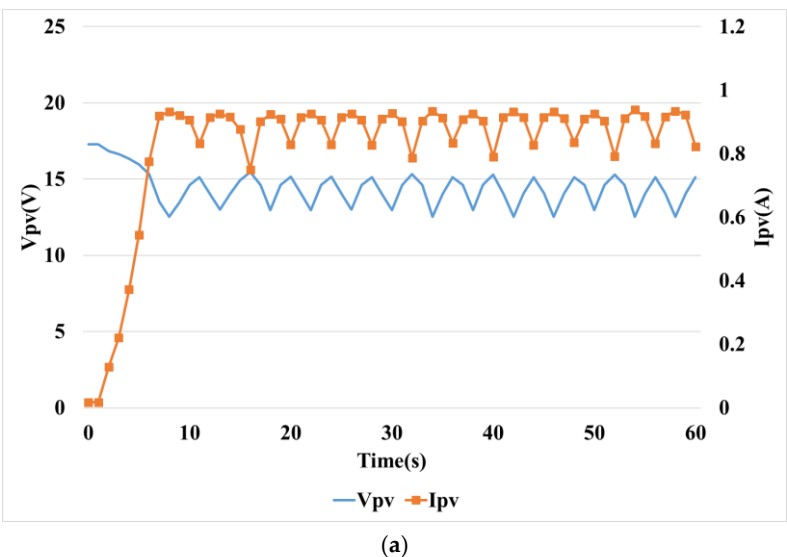

(**a**)

**Figure 16.** *Cont.*

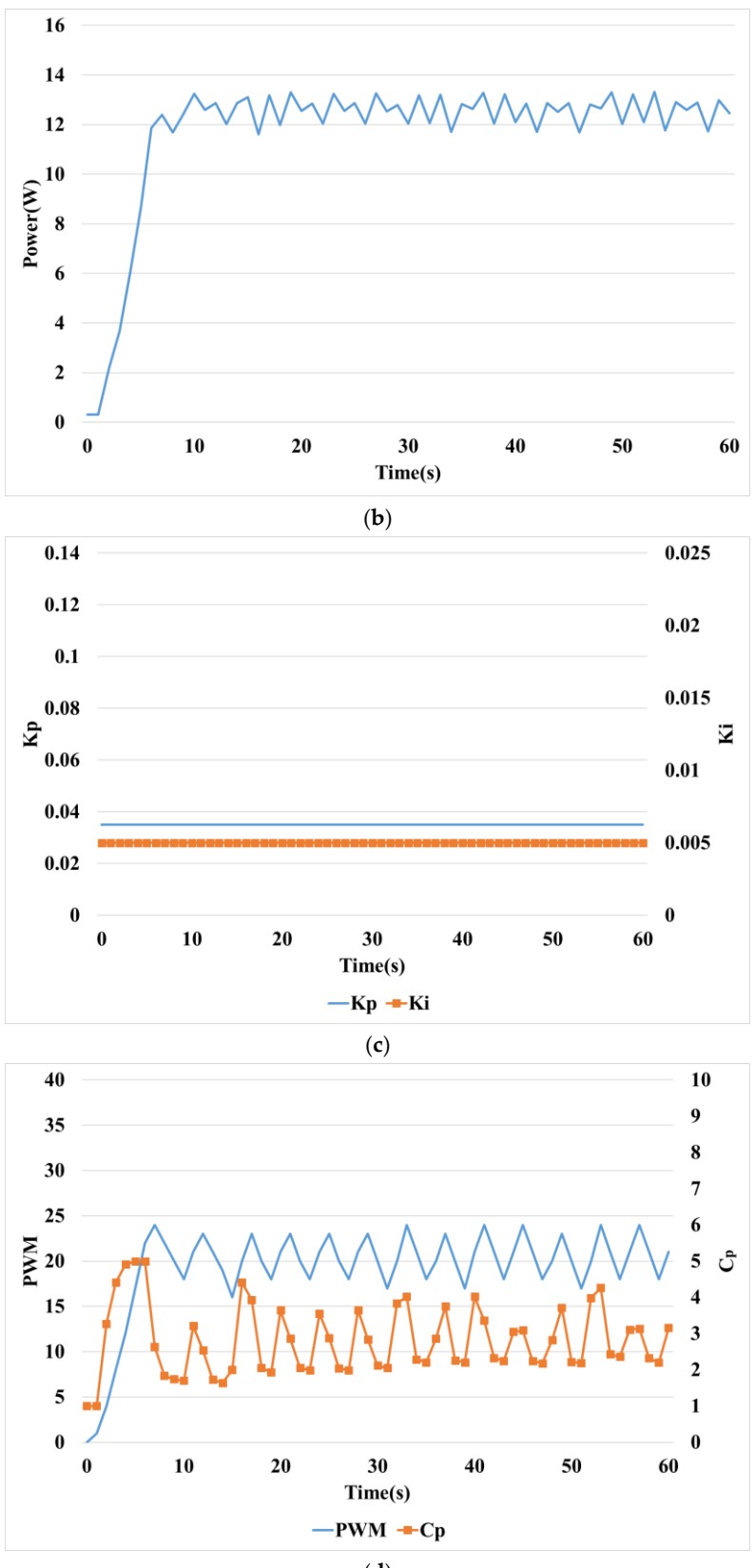

(**b**)

(**c**)

(**d**)

**Figure 16.** *Cont.*

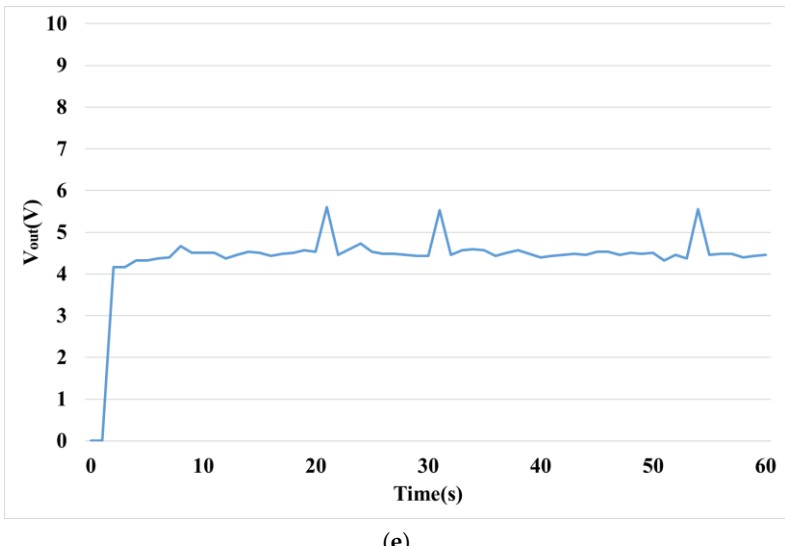

(**e**)

**Figure 16.** Response characteristics of the PI MPPT method. (**a**) Voltage ($V_{pv}$) and Current ($I_{pv}$) of the PV module; (**b**) Output Power of the PV module; (**c**) Proportional gain ($K_p$) and Integral gain ($K_i$) of the PI controller; (**d**) PWM signal and control value ($C_p$); (**e**) Output Voltage.

Figure 17 shows the response characteristics of the most commonly used P&O method for MPPT control. In particular, Figure 17a shows the voltage and current, Figure 17b presents the output power, and Figure 17c shows the PWM signal and control value ($C_p$). Since the P&O method uses the fixed control value ($C_p$) in both transient state and steady state, voltage in Figure 17a, power in Figure 17b, and PWM signal in Figure 17c have a constant ripple magnitude.

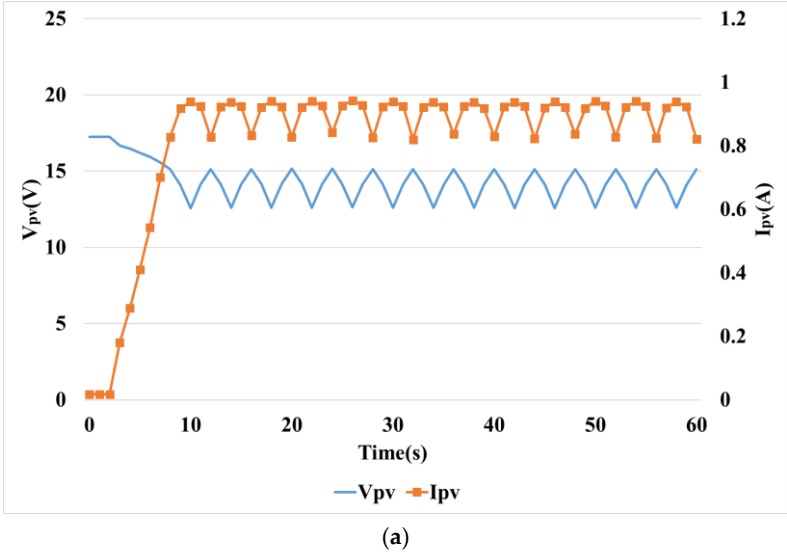

(**a**)

**Figure 17.** *Cont.*

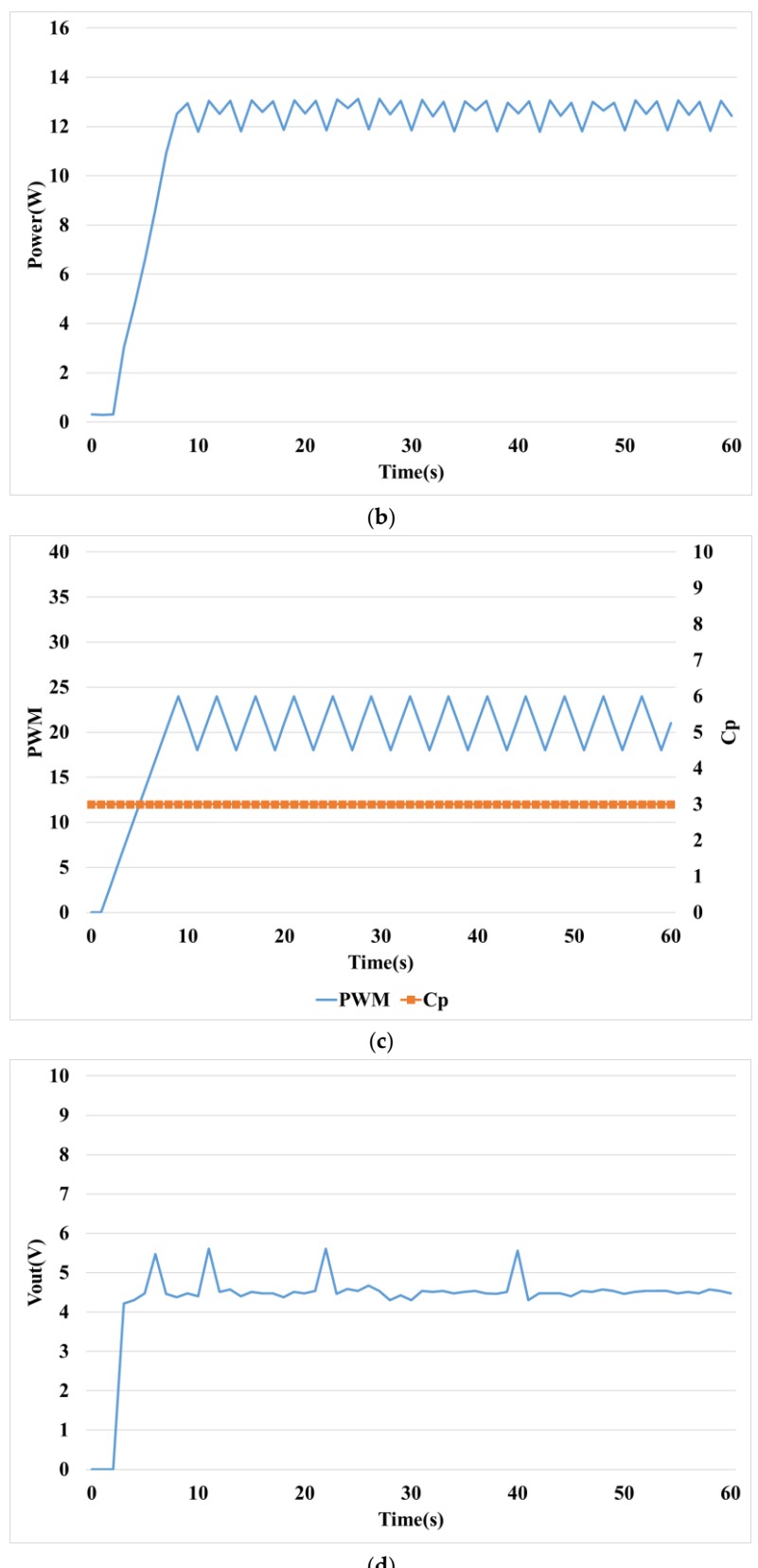

**Figure 17.** Response characteristics of the P&O MPPT method. (**a**) Voltage ($V_{pv}$) and Current ($I_{pv}$) of the PV module; (**b**) Output Power of the PV module; (**c**) PWM signal and control value ($C_P$); (**d**) Output Voltage.

Figure 18 and Table 6 show a comparison of the power response characteristics in transient states of the VGPI, PI, and P&O methods in Figures 15–17. The time to trace the maximum power point in transient state was measured as the time to reach the average power (12.5 [W]) of steady state, the results are shown in Table 6. As the results in Table 6 show, the VGPI controller has the most tracking speed with a high gain value in transient.

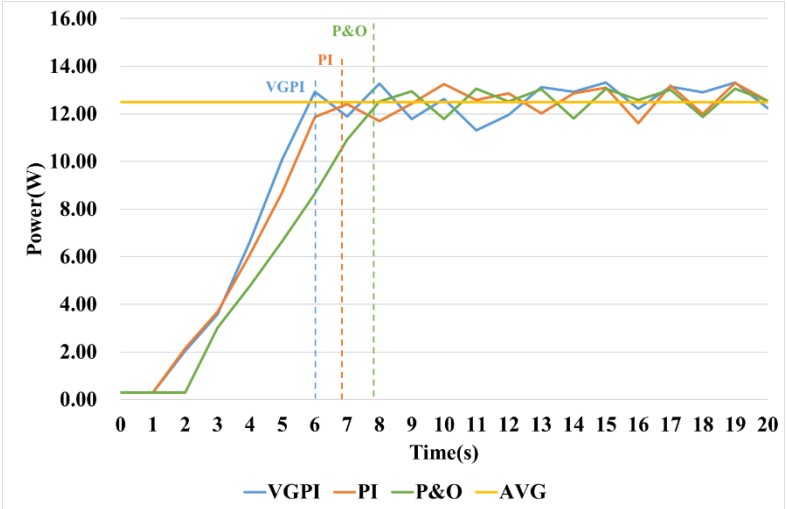

**Figure 18.** Comparison of response characteristics in transient state.

**Table 6.** Comparison of rising time in Figure 18.

|  | VGPI | PI | P&O |
|---|---|---|---|
| Rising Time(sec) | 5.94 (73.7%) | 7.13 (88.4%) | 8.06 (100%) |

Figure 19 and Table 7 represent the magnified picture and characteristics of the steady-state portion of Figures 15–17. The VGPI controller had the lowest error because it had lower gain values in a steady state, and the ripple was about 50% lower than the P&O method. Since the PI controller used a high gain value for fast tracking speed in transient conditions, steady-state error was rather higher than the P&O method.

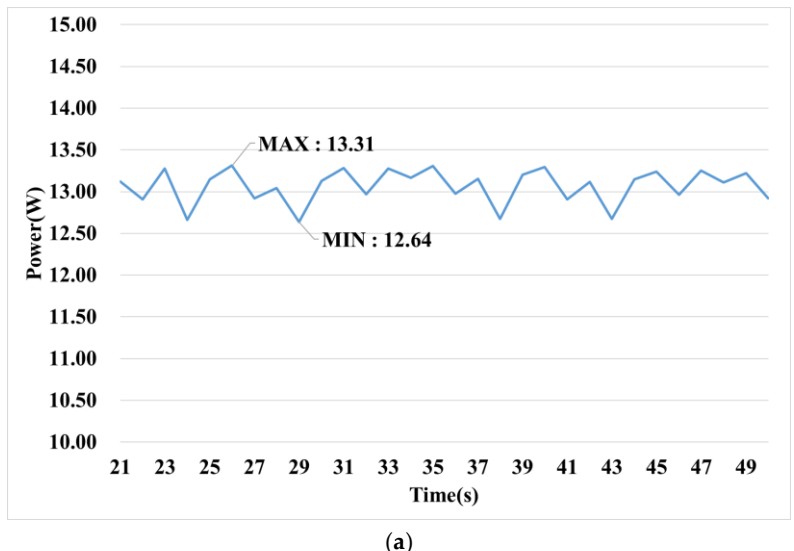

(a)

**Figure 19.** *Cont.*

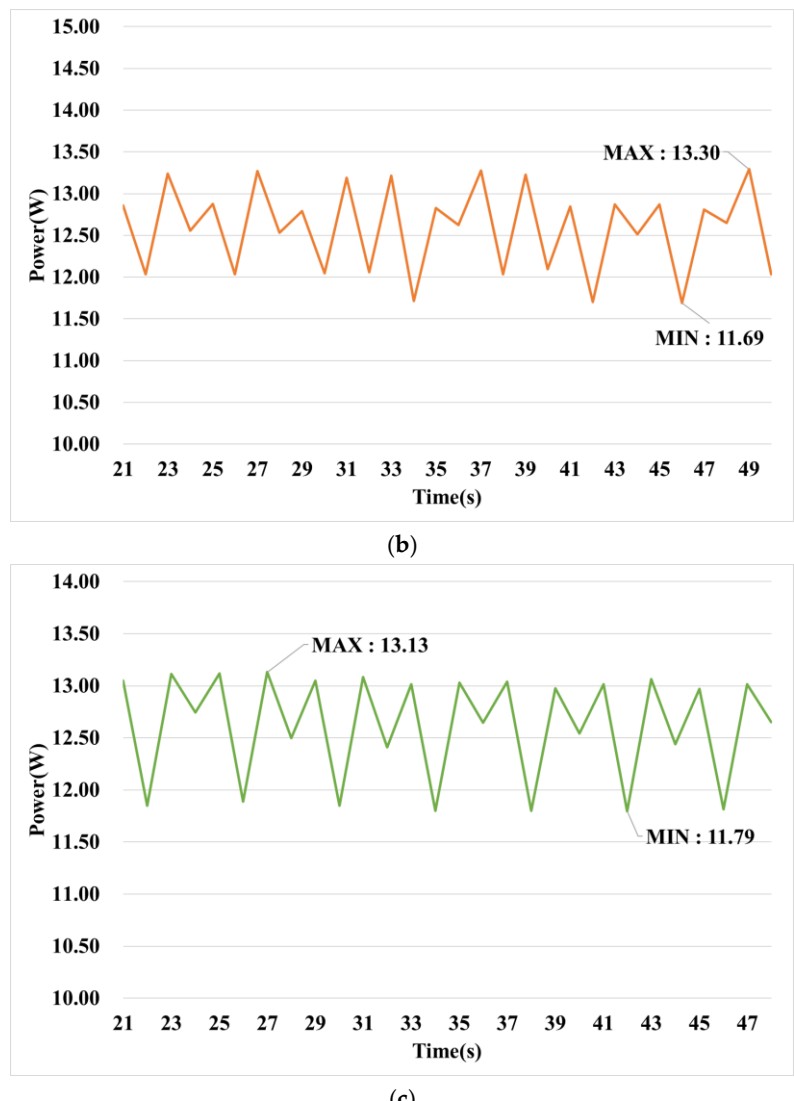

**Figure 19.** Comparison of response characteristics in steady state. (**a**) VGPI MPPT method; (**b**) PI MPPT method; (**c**) P&O MPPT method.

**Table 7.** Comparison of peak to peak in Figure 5.

|  | VGPI | PI | P&O |
|---|---|---|---|
| Min(W) | 12.64 | 11.69 | 11.79 |
| Max(W) | 13.31 | 13.30 | 13.13 |
| Peak to peak(W) | 0.67(50.3%) | 1.60(119.8%) | 1.13(100%) |

Figure 20 and Table 8 show MPPT control characteristics for changing conditions of solar radiation. Figure 20a represents the output of the VGPI, and Figure 20b represents the change in the gain value of the VGPI controller. The gain value of the VGPI controller represents a characteristic that is increasing in a transient state and is decreasing in steady state. Table 8 shows comparison of steady-state error for conditions with different solar irradiance. VGPI controllers show low steady-state error across all sections. Figure 20d represents the output voltage of the VGPI, PI, and P&O methods, with constant voltage output for the changing conditions of the solar radiation.

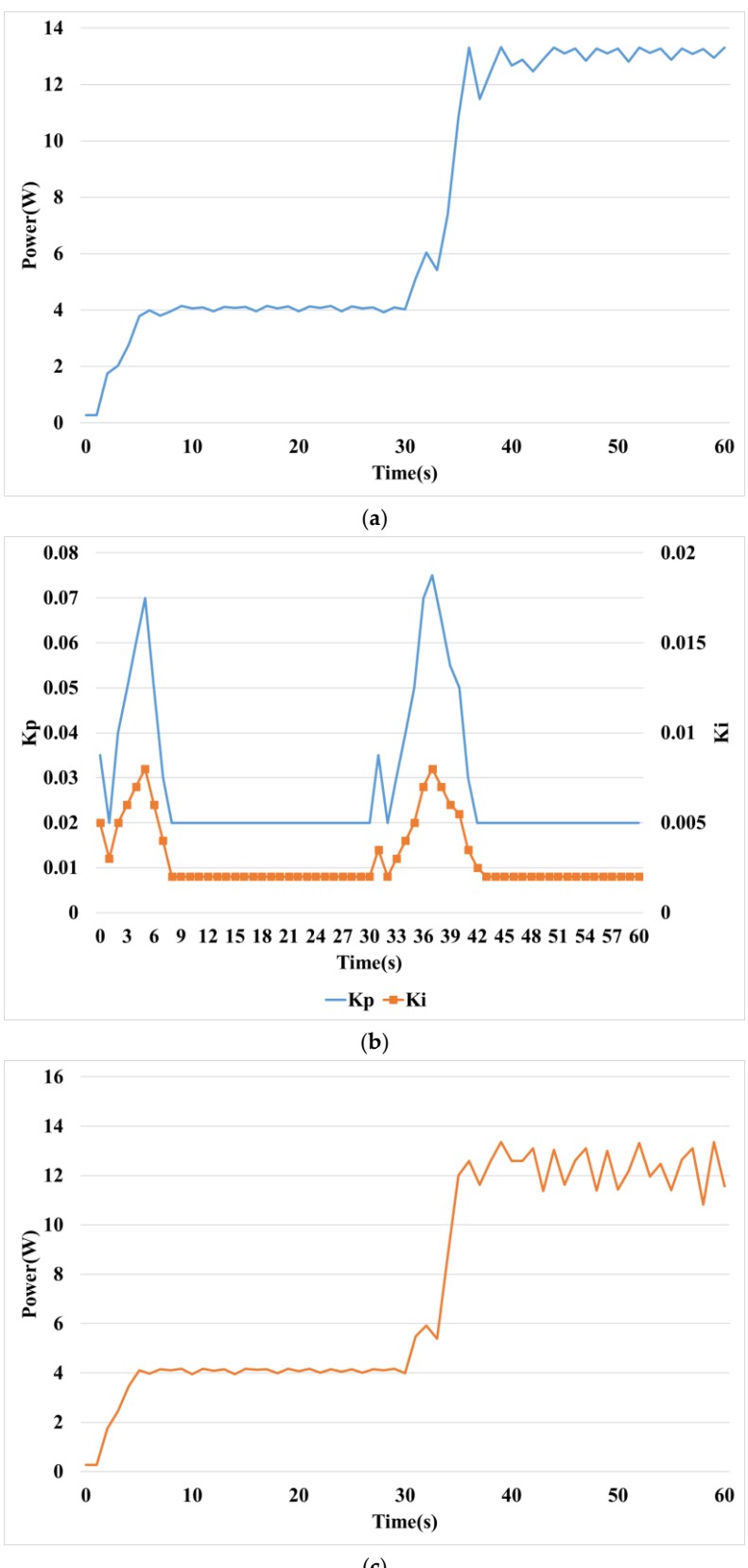

**Figure 20.** *Cont.*

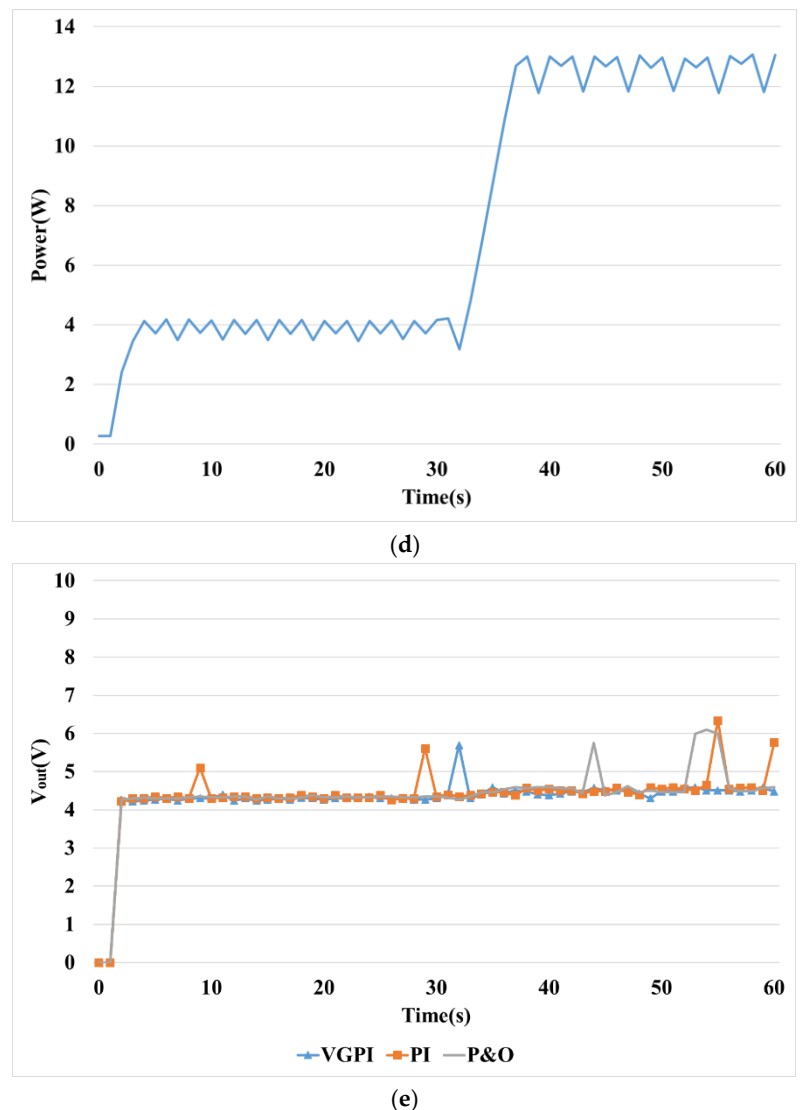

**Figure 20.** Comparison of MPPT control response characteristics for solar radiation variation. (**a**) Power of VGPI MPPT method; (**b**) Gain of VGPI method; (**c**) Power of PI MPPT method; (**d**) P&O MPPT method; (**e**) Output Voltage.

**Table 8.** Comparison of steady-state error to solar radiation variation.

|            |                 | VGPI          | PI             | P&O          |
|------------|-----------------|---------------|----------------|--------------|
| 11–30[sec] | MIN [W]         | 3.95          | 3.96           | 3.46         |
|            | MAX [W]         | 4.15          | 4.16           | 4.17         |
|            | Peak to Peak [W]| 0.20 (29.6%)  | 0.20 (29.6%)   | 0.71 (100%)  |
| 41–60[sec] | MIN [W]         | 12.47         | 10.82          | 11.78        |
|            | MAX [W]         | 13.31         | 13.35          | 13.07        |
|            | Peak to Peak [W]| 0.84 (65.0%)  | 2.53 (195.9%)  | 1.29 (100%)  |

## 5. Conclusions

This paper proposes a method for tracking the MPP of solar power generation. P&O and IncCond methods, which are commonly used MPPT methods, have limitations in performance improvement because they track a MPP by varying a voltage or a current with a constant magnitude.

In this paper, the VGPI controller is proposed to solve this problem. The characteristics of the VGPI controller presented in the paper are as follows.

1. Based on the PI controller, which is most commonly used in industrial sites.
2. The gain value of the PI controller adjusts according to the operating state using fuzzy control.
3. Design a fuzzy control membership function and rule base in accordance with the basic operation of the PV module.
4. The membership function of the fuzzy control output consists of three types: increase (P: Positive), hold (ZE: zero) and decrease (N: negative).

The gain value of the PI controller is increased for a quick response in transient conditions and reduced to reduce steady-state error in normal conditions. The VGPI proposed in this paper compares the tracking time at transient-state and the error in steady state with conventional MPPT methods for two scenarios (constant and changing conditions of solar radiation).

Under constant or varying conditions, the proportional gain ($K_p$) and integral gain ($K_i$) of the VGPI controller increased to the maximum value in the transient state and decreased to the minimum value in the steady state by the fuzzy control, and were continuously adjusted.

As a result, the VGPI controller proposed in this paper is about 14% better than the conventional PI and P&O in tracking speed, and the error in steady-state shown respectively 36.5% and 40% lower than PI and P&O. Even in conditions with varying solar radiation, the VGPI controller had excellent MPPT performance than other controllers because of continuous gain adjustment. VGPI controllers are able to adjust continuous gain values according to changing environments, and both transient and steady-state response performance was improved.

This method is expected to be applicable to various variable systems, as well as MPPT for solar power. Fuzzy control used in this paper requires continuous calculation according to changing environmental conditions. In addition, the calculation of fuzzy control depends on the membership function and rule base, and the calculation amount can be greatly increased according to the environmental change conditions. This phenomenon degrades the MPPT performance.

**Author Contributions:** Conceptualization, J.-C.K. and J.-S.K.; Data curation, J.-C.K. and J.-S.K.; Formal analysis, J.-C.K. and J.-H.H.; Funding acquisition, J.-H.H.; Investigation, J.-H.H.; Methodology, J.-C.K., J.-H.H. and J.-S.K.; Project administration, J.-H.H.; Software, J.-H.H. and J.-S.K.; Visualization, J.-S.K.; Writing—original draft, J.-C.K., J.-H.H. and J.-S.K.; Writing—review & editing, J.-H.H. and J.-S.K.

**Funding:** This research was supported by Energy Cloud R&D Program through the National Research Foundation of Korea (NRF) funded by the Ministry of Science, ICT (NRF-2019M3F2A1073385).

**Conflicts of Interest:** The authors declare no conflict of interest.

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
