# Peer review of "Improvement of MPPT Control Performance Using Fuzzy Control and VGPI in the PV System for Micro Grid"

_sustainability, doi:10.3390/su11215891_

Round 1

Reviewer 1 Report

Dear authors,

Your paper propose an interesting methodology that improves the MPPT control performance using Fuzzy control and VGPI in the PV Systems.

1. Please, review the units according the International System of Units. i.e. line 101 "w/m^2" instead of "W/m^2"; line 104 Coulombs "c" instead of "C"; ...  

2. From my modest point of view the paper is unnecesary long. For instance, Section 2 provides the basic equations of a PV cell. This information can be found in almost any textbook. The same happens in Section 3 with the description of the basic behavior of a Buck converter.

3. The set-up is properly described but it is not clear for me the criteria followed to define the scenarios under study. Please, provide more details. From my modets point of view this is the weakest point of the paper.

4. It could be useful for the readers to know what are the performance values of other MPPT methods operating in similar conditions. That will help the readers to understand why this method can exhibit better performance that other methods.

5. Please, try to define in a more clear way what is the novelty of this paper

Thank you very much for your contribution

Best Regards

Author Response

Does the introduction provide sufficient background and include all relevant references?
 (x) ( ) ( ) ( )
Is the research design appropriate?
( ) (x) ( ) ( )
Are the methods adequately described?
( ) (x) ( ) ( )
Are the results clearly presented?
( ) ( ) (x) ( )

-----
Reply-----

We have added the following to make the results clearer.
ADD 1)
The PI controller used for comparison with VGPI uses 0.055 for proportional gain(K_p) and 0.007 for integral gain(K_i). Arduino's PWM ranges from 0 to 255, with 0 representing 0% and 255 representing 100% duty ratio. The P&O controller adjusts the PWM to a fixed size 3 to regulate the voltage at a constant rate.
ADD 2)
Figure 20(b) and 20(d) show the change in the gain value of the VGPI and PI controllers respectively. The gain value of the VGPI controller shows characteristic which gains are increasing in transient-state and decreasing until reaching a minimum value in steady-state.

(b)

(d)
(a) Power of VGPI MPPT method; (b) Gain of VGPI method; (c) Power of PI MPPT method; (d) Gain of PI method; (e) P&O MPPT method
Figure 20. Comparison of MPPT control response characteristics for solar radiation variation

Are the conclusions supported by the results?
( ) (x) ( ) ( )
Comments and Suggestions for Authors

Dear authors,
Your paper propose an interesting methodology that improves the MPPT control performance using Fuzzy control and VGPI in the PV Systems.
1. Please, review the units according the International System of Units. i.e. line 101 "w/m^2" instead of "W/m^2"; line 104 Coulombs "c" instead of "C"; ... 

-----

Reply-----

We removed Chapter 2 based on the opinion of reviewer that the contents of the paper are too long.
REMOVE 1)
2. Solar Cell Modeling
A solar cells consist of one ideal diode and a constant current source (I_ph). In reality, however, it is impossible to make an ideal diode; thus, a series resistor (R_s) and a parallel resistor (R_ph) representing the contact resistance and sheet resistance, respectively, of the surface layer must be considered. A part of the light incident on the surface of the solar cell is reflected from the surface, the light transmitted through the surface is absorbed in the solar cell, and the number of photons decreases exponentially.
Figure 1 shows the equivalent circuit of a solar cell.

Figure 1. Equivalent of solar cell

 The photocurrent (I_ph) is proportional to the solar radiation and is given by the following equation:
I_ph=(G/G_0 ) I_g0+J_0 (T_c-T_ref) (1)

Where G_o is the reference solar irradiance, G is the solar irradiance that enters the solar cell. I_g0 is the current at the reference solar irradiance (G_0), J_o is the temperature coefficient for photocurrent (I_ph), T_c is the temperature of the cell, and T_ref is the reference temperature of the cell. The reference radiation used in the equation representing photocurrent is mostly 1000[W/m^2]. The diode current (I_d) as shown in Figure 1 is given by the Shockley equation as follows.
I_d=I_0 [e^((q(v_pv+I_pv R_s )/nkT_c ) )-1] (2)
Here, V_pv+I_pv R_s represents the voltage of the diode, I_0 denotes the diode inverse saturation current, and q refers to the amount of electrons [q=1.602×〖10〗^(-19)  [C]]. V_pv and I_pv are the cell voltage and current, respectively, R_s is the series resistance, n is the ideal coefficient, and k is the Boltzmann constant (k=1.38×〖10〗^(-23)  [J/K]). Diode reverse saturation current I_0 is temperature-sensitive and can be expressed as:
I_0=I_d0 (T_c/T_ref )^3 e^[qE_g/nk (1/T_ref -1/T_c )]  (3)
Where I_d0 represents the diode inverse saturation current at the reference temperature, and T_ref  and T_c use the Kelvin temperature. Bandgap energy E_g of the silicon semiconductor constituting the solar cell can be expressed by the following equation:

E_g=1.17-(4.73×〖10〗^(-4)×T_c^2)/(T_c+636) (4)
The temperature of the solar cell (T_c) is proportional to the amount of solar radiation and can be expressed as:
T_c=273+T_a+((NOCT-20)/800)×G (5)
Where T_a represents the atmospheric temperature (℃) and NOCT (Nominal Operating Cell Temperature) denotes the nominal solar cell operating temperature. In the solar cell equivalent, circuit shown in Figure 1, the I_pv can be shown as follows.
I_pv=I_ph-I_o [e^(q(V_pv+I_pv×R_s )/(nkT_c ))-1]-(〖V_pv+I〗_pv R_s)/R_sh  (6)
Where R_sh represents the parallel resistance. The current in Equation (6) is common to the left and right equations, and the relationship of I_pv-V_pv can be expressed as [3, 35-38]:
f(I_pv,V_pv,G)=I_pv-{I_ph (G)-I_0 (G)[e^((q(V_pv+I_pv R_s )/(nkT_c (G) )) )-1]-(V_pv+(I_pv R_s)/R_sh )}=0 (7)

2. From my modest point of view the paper is unnecesary long. For instance, Section 2 provides the basic equations of a PV cell. This information can be found in almost any textbook. The same happens in Section 3 with the description of the basic behavior of a Buck converter.

-----

Reply-----

We have removed unnecessary parts from the paper according to the reviewer's opinion.

REMOVE 1)
Fossil fuels have a problem of resource financing and environmental pollution, and interest in alternative energy sources is on the rise. In particular, greenhouse gases emitted by fossil fuels account for more than 80% of the total. Oil, natural gas, and coal are expected to be depleted due to the rising energy demand [1–2].

REMOVE 2)
2. Solar Cell Modeling
A solar cells consist of one ideal diode and a constant current source (I_ph). In reality, however, it is impossible to make an ideal diode; thus, a series resistor (R_s) and a parallel resistor (R_ph) representing the contact resistance and sheet resistance, respectively, of the surface layer must be considered. A part of the light incident on the surface of the solar cell is reflected from the surface, the light transmitted through the surface is absorbed in the solar cell, and the number of photons decreases exponentially.
Figure 1 shows the equivalent circuit of a solar cell.

Figure 1. Equivalent of solar cell

 The photocurrent (I_ph) is proportional to the solar radiation and is given by the following equation:
I_ph=(G/G_0 ) I_g0+J_0 (T_c-T_ref) (2)

Where G_o is the reference solar irradiance, G is the solar irradiance that enters the solar cell. I_g0 is the current at the reference solar irradiance (G_0), J_o is the temperature coefficient for photocurrent (I_ph), T_c is the temperature of the cell, and T_ref is the reference temperature of the cell. The reference radiation used in the equation representing photocurrent is mostly 1000[W/m^2]. The diode current (I_d) as shown in Figure 1 is given by the Shockley equation as follows.
I_d=I_0 [e^((q(v_pv+I_pv R_s )/nkT_c ) )-1] (2)
Here, V_pv+I_pv R_s represents the voltage of the diode, I_0 denotes the diode inverse saturation current, and q refers to the amount of electrons [q=1.602×〖10〗^(-19)  [C]]. V_pv and I_pv are the cell voltage and current, respectively, R_s is the series resistance, n is the ideal coefficient, and k is the Boltzmann constant (k=1.38×〖10〗^(-23)  [J/K]). Diode reverse saturation current I_0 is temperature-sensitive and can be expressed as:
I_0=I_d0 (T_c/T_ref )^3 e^[qE_g/nk (1/T_ref -1/T_c )]  (3)
Where I_d0 represents the diode inverse saturation current at the reference temperature, and T_ref  and T_c use the Kelvin temperature. Bandgap energy E_g of the silicon semiconductor constituting the solar cell can be expressed by the following equation:

E_g=1.17-(4.73×〖10〗^(-4)×T_c^2)/(T_c+636) (4)
The temperature of the solar cell (T_c) is proportional to the amount of solar radiation and can be expressed as:
T_c=273+T_a+((NOCT-20)/800)×G (5)
Where T_a represents the atmospheric temperature (℃) and NOCT (Nominal Operating Cell Temperature) denotes the nominal solar cell operating temperature. In the solar cell equivalent, circuit shown in Figure 1, the I_pv can be shown as follows.
I_pv=I_ph-I_o [e^(q(V_pv+I_pv×R_s )/(nkT_c ))-1]-(〖V_pv+I〗_pv R_s)/R_sh  (6)
Where R_sh represents the parallel resistance. The current in Equation (6) is common to the left and right equations, and the relationship of I_pv-V_pv can be expressed as [3, 35-38]:
f(I_pv,V_pv,G)=I_pv-{I_ph (G)-I_0 (G)[e^((q(V_pv+I_pv R_s )/(nkT_c (G) )) )-1]-(V_pv+(I_pv R_s)/R_sh )}=0 (7)
REMOVE 3)

(b)

(c)
(a) Buck converter; (b) On-state; (c) Off-state
Figure 3. Structure of the buck converter

3. The set-up is properly described but it is not clear for me the criteria followed to define the scenarios under study. Please, provide more details. From my modets point of view this is the weakest point of the paper.

-----
Reply-----

We added the following information about the scenario in the paper.
ADD 1)
Methods for adjusting the gain value of PI controller using fuzzy control were presented through several studies[34-38]. However, existing studies depend on designer knowledge for rule base and membership function designs and do not suggest how the gain value of a PI controller changes with its operational state. Thus, the paper proposes simple and clear fuzzy control membership function and rule base design according to the characteristics of the PV system and shows the gains of PI controller which are changing in the transient and steady-state of the system.

ADD2)
Various methods of adjusting the gain value of the PI controller using fuzzy control have been proposed. These methods are based on user knowledge in designing Fuzzy Control's membership functions and rule base and do not represent the background to design. This approach has the problem of redesigning membership functions and rule bases for other users to use.
Therefore, in this paper, using the operating characteristics for the gain value of the PI controller, a simpler and easier-to-understand controller is designed. Member functions and rule bases designed in the paper are based on the following:

 Error and changing error which is the input of fuzzy control, are as shown in expressions (19) and (20).
 The error and the changing error decrease in size as solar power is closer to the MPP.
 If the input value is large, the tracking speed should be fast because it is far from the MPP. This increases the gain value of the PI controller.
 If the input value is small, it is close to the MPP and the error in steady state must be reduced. This reduces the gain value of the PI controller.

The input of fuzzy control, error and error variation, are divided into seven sections: Negative big(NB), Negative medium(NM), Negative small(NS), zero(ZE), Positive big(PB), Positive medium(PM) and positive small(PS). The output of the fuzzy control is designed to perform three actions: increase(P: positive), hold(ZE: zero) and decrease(N: negative).
Tables 4 and 5 show the rule base for proportional gain (K_p) and integral gain (K_i) designed in the paper, and Figures 9–11 show membership function for the input and output of fuzzy control.

4. It could be useful for the readers to know what are the performance values of other MPPT methods operating in similar conditions. That will help the readers to understand why this method can exhibit better performance that other methods.

-----

Reply-----

We added control values of the PI controller and P & O method used to compare with the VGPI controller presented in the paper.

ADD 1)
The PI controller used for comparison with VGPI uses 0.055 for proportional gain(K_p) and 0.007 for integral gain(K_i). Arduino's PWM ranges from 0 to 255, with 0 representing 0% and 255 representing 100% duty ratio. The P&O controller adjusts the PWM to a fixed size 3 to regulate the voltage at a constant rate.

5. Please, try to define in a more clear way what is the novelty of this paper
Thank you very much for your contribution
Best Regards

-----

Reply-----

We have revised the conclusions as follows to represent the novelty presented in the paper.

CHANGE 1)
This paper proposes a method for tracking the MPP of solar power generation. P&O and IncCond methods, which are commonly used MPPT methods, have limitations in performance improvement because they track a MPP by varying a voltage or a current with a constant magnitude.
In this paper, the VGPI controller is proposed to solve this problem. The characteristics of the VGPI controller presented in the paper are as follows.

 Based on the PI controller, which is most commonly used in industrial sites.
 The gain value of the PI controller adjusts according to the operating state using fuzzy control.
 Design a fuzzy control membership function and rule base in accordance with the basic operation of the PV module.
 The membership function of the fuzzy control output consists of three types: increase (P: Positive), hold (ZE : zero) and decrease(N: negative)

The gain value of the PI controller is increased for quick response in transient conditions and reduced to reduce steady-state error in normal conditions. The VGPI proposed in this paper compares the tracking time at transient-state and the error in steady state with conventional MPPT methods for two scenarios (constant and changing conditions of solar radiation).
Under constant or varying conditions, the proportional gain (K_p) and integral gain (K_i) of the VGPI controller increased to the maximum value in the transient state and decreased to the minimum value in the steady state by the fuzzy control, and were continuously adjusted.
As a result, the VGPI controller proposed in this paper is about 14% better than the conventional PI and P&O in tracking speed, and the error in steady-state shown respectively 36.5% and 40% lower than PI and P&O. Even in conditions with varying solar radiation, the VGPI controller had excellent MPPT performance than other controllers because of continuous gain adjustment. VGPI controllers are able to adjust continuous gain values according to changing environments, and both transient and steady-state response performance was improved.
This method is expected to be applicable to various variable systems, as well as MPPT for solar power. Fuzzy control used in this paper requires continuous calculation according to changing environmental conditions. And the calculation of fuzzy control depends on the membership function and rule base, and the calculation amount can be greatly increased according to the environmental change conditions. This phenomenon degrades the MPPT performance.

Reviewer 2 Report

This paper presented a MPPT method for PV applications with VGPI controller.

The writing of the paper is ok, However, the novelty of the paper is low. Similar work has been published in many papers, and very little new contribution can be found from this paper. There are too much basic things in the paper, and which can be found in textbooks, such as the working principle of buck converters.

Author Response

Are the conclusions supported by the results?
( ) ( ) (x) ( )

-----

Reply-----

We have revised the conclusion so that the conclusion is supported by the results.

CHANGE 1)
This paper proposes a method for tracking the MPP of solar power generation. P&O and IncCond methods, which are commonly used MPPT methods, have limitations in performance improvement because they track a MPP by varying a voltage or a current with a constant magnitude.
In this paper, the VGPI controller is proposed to solve this problem. The characteristics of the VGPI controller presented in the paper are as follows.

 Based on the PI controller, which is most commonly used in industrial sites.
 The gain value of the PI controller adjusts according to the operating state using fuzzy control.
 Design a fuzzy control membership function and rule base in accordance with the basic operation of the PV module.
 The membership function of the fuzzy control output consists of three types: increase (P: Positive), hold (ZE : zero) and decrease(N: negative)

The gain value of the PI controller is increased for quick response in transient conditions and reduced to reduce steady-state error in normal conditions. The VGPI proposed in this paper compares the tracking time at transient-state and the error in steady state with conventional MPPT methods for two scenarios (constant and changing conditions of solar radiation).
Under constant or varying conditions, the proportional gain (K_p) and integral gain (K_i) of the VGPI controller increased to the maximum value in the transient state and decreased to the minimum value in the steady state by the fuzzy control, and were continuously adjusted.
As a result, the VGPI controller proposed in this paper is about 14% better than the conventional PI and P&O in tracking speed, and the error in steady-state shown respectively 36.5% and 40% lower than PI and P&O. Even in conditions with varying solar radiation, the VGPI controller had excellent MPPT performance than other controllers because of continuous gain adjustment. VGPI controllers are able to adjust continuous gain values according to changing environments, and both transient and steady-state response performance was improved.
This method is expected to be applicable to various variable systems, as well as MPPT for solar power. Fuzzy control used in this paper requires continuous calculation according to changing environmental conditions. And the calculation of fuzzy control depends on the membership function and rule base, and the calculation amount can be greatly increased according to the environmental change conditions. This phenomenon degrades the MPPT performance.

Comments and Suggestions for Authors
This paper presented a MPPT method for PV applications with VGPI controller.
The writing of the paper is ok, However, the novelty of the paper is low. Similar work has been published in many papers, and very little new contribution can be found from this paper. There are too much basic things in the paper, and which can be found in textbooks, such as the working principle of buck converters.

-----

Reply-----

We have reduced too many basic points used in the paper.

CHANGE 1)
Fossil fuels have a problem of resource financing and environmental pollution, and interest in alternative energy sources is on the rise. In particular, greenhouse gases emitted by fossil fuels account for more than 80% of the total. Oil, natural gas, and coal are expected to be depleted due to the rising energy demand [1–2].

CHANGE 2)
2. Solar Cell Modeling
A solar cells consist of one ideal diode and a constant current source (I_ph). In reality, however, it is impossible to make an ideal diode; thus, a series resistor (R_s) and a parallel resistor (R_ph) representing the contact resistance and sheet resistance, respectively, of the surface layer must be considered. A part of the light incident on the surface of the solar cell is reflected from the surface, the light transmitted through the surface is absorbed in the solar cell, and the number of photons decreases exponentially.
Figure 1 shows the equivalent circuit of a solar cell.

Figure 1. Equivalent of solar cell

 The photocurrent (I_ph) is proportional to the solar radiation and is given by the following equation:
I_ph=(G/G_0 ) I_g0+J_0 (T_c-T_ref) (3)

Where G_o is the reference solar irradiance, G is the solar irradiance that enters the solar cell. I_g0 is the current at the reference solar irradiance (G_0), J_o is the temperature coefficient for photocurrent (I_ph), T_c is the temperature of the cell, and T_ref is the reference temperature of the cell. The reference radiation used in the equation representing photocurrent is mostly 1000[W/m^2]. The diode current (I_d) as shown in Figure 1 is given by the Shockley equation as follows.
I_d=I_0 [e^((q(v_pv+I_pv R_s )/nkT_c ) )-1] (2)
Here, V_pv+I_pv R_s represents the voltage of the diode, I_0 denotes the diode inverse saturation current, and q refers to the amount of electrons [q=1.602×〖10〗^(-19)  [C]]. V_pv and I_pv are the cell voltage and current, respectively, R_s is the series resistance, n is the ideal coefficient, and k is the Boltzmann constant (k=1.38×〖10〗^(-23)  [J/K]). Diode reverse saturation current I_0 is temperature-sensitive and can be expressed as:
I_0=I_d0 (T_c/T_ref )^3 e^[qE_g/nk (1/T_ref -1/T_c )]  (3)
Where I_d0 represents the diode inverse saturation current at the reference temperature, and T_ref  and T_c use the Kelvin temperature. Bandgap energy E_g of the silicon semiconductor constituting the solar cell can be expressed by the following equation:

E_g=1.17-(4.73×〖10〗^(-4)×T_c^2)/(T_c+636) (4)
The temperature of the solar cell (T_c) is proportional to the amount of solar radiation and can be expressed as:
T_c=273+T_a+((NOCT-20)/800)×G (5)
Where T_a represents the atmospheric temperature (℃) and NOCT (Nominal Operating Cell Temperature) denotes the nominal solar cell operating temperature. In the solar cell equivalent, circuit shown in Figure 1, the I_pv can be shown as follows.
I_pv=I_ph-I_o [e^(q(V_pv+I_pv×R_s )/(nkT_c ))-1]-(〖V_pv+I〗_pv R_s)/R_sh  (6)
Where R_sh represents the parallel resistance. The current in Equation (6) is common to the left and right equations, and the relationship of I_pv-V_pv can be expressed as [3, 35-38]:
f(I_pv,V_pv,G)=I_pv-{I_ph (G)-I_0 (G)[e^((q(V_pv+I_pv R_s )/(nkT_c (G) )) )-1]-(V_pv+(I_pv R_s)/R_sh )}=0 (7)

CHANGE 3)

(b)

(c)
(a) Buck converter; (b) On-state; (c) Off-state
Figure 3. Structure of the buck converter

-----

Reply-----

We have added the following to explain the differences between the presented and the other papers.
ADD 1)
Methods for adjusting the gain value of PI controller using fuzzy control were presented through several studies[34-38]. However, existing studies depend on designer knowledge for rule base and membership function designs and do not suggest how the gain value of a PI controller changes with its operational state. Thus, the paper proposes simple and clear fuzzy control membership function and rule base design according to the characteristics of the PV system and shows the gains of PI controller which are changing in the transient and steady-state of the system.

ADD 2)
 Pal, A. K.; Mudi, R. K. Self-Tuning Fuzzy PI Controller and its Application to HVAC System. International journal of computational cognition(IJCC). 2008, 1, 25-30.
 Kassem, A. M. Fuzzy-logic Based Self-tuning PI Controller for High-Performance Vector Controlled Induction Motor Fed by PV-Generator. WSEAS TRANSACTIONS on SYSTEMS. 2013, 12, 22-31.
 Wahyunggoro, O.; Saad, N. Development of Fuzzy-logic-based Self Tuning PI Controller for Servomotor. Advanced Strategies for Robot Manipulators. 2010, 311-328
 Anantwar, H.; Lakshmikantha, B. R.; Sundar, S. Fuzzy self tuning PI controller based inverter control for voltage regulation in off-grid hybrid power system. International Conference on Power Engineering, Computing and CONtrol(PECCON). 2017, 117, 409-416.
 Mudi, R. K.; Pal, N. R. A self-tuning fuzzy PI controller. Fuzzy sets and Systems. 2000, 115, 327-338.

CHANGE 1)
5. Conclusion
This paper proposes a method for tracking the MPP of solar power generation. P&O and IncCond methods, which are commonly used MPPT methods, have limitations in performance improvement because they track a MPP by varying a voltage or a current with a constant magnitude.
In this paper, the VGPI controller is proposed to solve this problem. The characteristics of the VGPI controller presented in the paper are as follows.

 Based on the PI controller, which is most commonly used in industrial sites.
 The gain value of the PI controller adjusts according to the operating state using fuzzy control.
 Design a fuzzy control membership function and rule base in accordance with the basic operation of the PV module.
 The membership function of the fuzzy control output consists of three types: increase (P: Positive), hold (ZE : zero) and decrease(N: negative)

The gain value of the PI controller is increased for quick response in transient conditions and reduced to reduce steady-state error in normal conditions. The VGPI proposed in this paper compares the tracking time at transient-state and the error in steady state with conventional MPPT methods for two scenarios (constant and changing conditions of solar radiation).
Under constant or varying conditions, the proportional gain (K_p) and integral gain (K_i) of the VGPI controller increased to the maximum value in the transient state and decreased to the minimum value in the steady state by the fuzzy control, and were continuously adjusted.
As a result, the VGPI controller proposed in this paper is about 14% better than the conventional PI and P&O in tracking speed, and the error in steady-state shown respectively 36.5% and 40% lower than PI and P&O. Even in conditions with varying solar radiation, the VGPI controller had excellent MPPT performance than other controllers because of continuous gain adjustment. VGPI controllers are able to adjust continuous gain values according to changing environments, and both transient and steady-state response performance was improved.
This method is expected to be applicable to various variable systems, as well as MPPT for solar power. Fuzzy control used in this paper requires continuous calculation according to changing environmental conditions. And the calculation of fuzzy control depends on the membership function and rule base, and the calculation amount can be greatly increased according to the environmental change conditions. This phenomenon degrades the MPPT performance.

-----
Reply-----

We have added a change in the gain value of the PI controller for clarity and differentiation of the controller presented in the results.

ADD 1)
Figure 20(b) and 20(d) show the change in the gain value of the VGPI and PI controllers respectively. The gain value of the VGPI controller shows characteristic which gains are increasing in transient-state and decreasing until reaching a minimum value in steady-state.

ADD 2)

(b)

(d)
 (a) Power of VGPI MPPT method; (b) Gain of VGPI method; (c) Power of PI MPPT method; (d) Gain of PI method; (e) P&O MPPT method
Figure 20. Comparison of MPPT control response characteristics for solar radiation variation

Reviewer 3 Report

Review Comments on sustainability-583644

Improvement of MPPT Control Performance Using Fuzzy Control and VGPI in the PV System for Microgrid

The abstract needs to be rephrased. Please take note of the sequence in terms of technical development for MPPT. First, PI controller is mentioned and its shortcomings are mentioned. In order to overcome the disadvantage of conventional PI controller, a new fuzzy PI controller is proposed in this paper. ….. On Page 1, delete the first 4 sentences from Line 29 to 33. Use only one sentence to mention application of solar energy and its development trend. Since the acronym of maximum power point tracking is given as MPPT at the beginning of this paper. MPPT should be applied through the text. It is not appropriate that maximum power point tracking is used sometime and MPPT is used some other times. Where maximum power point tracking appears, it must be replaced by MPPT through the paper. Line 48 on Page 2, remove ‘to control’. Line 85 on Page 2, it should be ‘ A solar cell consists of …’. Line 101 on Page 3, it should be ‘The diode current (Id) as shown in Figure 1 is given by the Shockley equation as follows: ‘. Line 104 on Page 3, what does [c] mean? After an equation, a stop ‘.’ must be added as a new sentence follows. In equation (1), G is not defined. Lines 115 ~ 116, “The relationship of … “ must be rephrased as the meaning is not clear. Flow charts shown in Figs. 6 and 8 should be associated with citation numbers if the charts are borrowed from the published articles. The conclusion section must be revised as the current writing is not a conclusion. It is rather written as an introduction. In conclusion, the authors need to give what have been achieved or obtained in this study, what the innovative approaches have been developed. A general overview or introduction to the domain knowledge must be removed.   It is not sure if the results presented are all correct. The authors must conduct a careful check again. 

Author Response

Are the conclusions supported by the results?
( ) ( ) (x) ( )

-----

Reply-----

We have revised the conclusion so that the conclusion is supported by the results.

CHANGE 1)
This paper proposes a method for tracking the MPP of solar power generation. P&O and IncCond methods, which are commonly used MPPT methods, have limitations in performance improvement because they track a MPP by varying a voltage or a current with a constant magnitude.
In this paper, the VGPI controller is proposed to solve this problem. The characteristics of the VGPI controller presented in the paper are as follows.

 Based on the PI controller, which is most commonly used in industrial sites.
 The gain value of the PI controller adjusts according to the operating state using fuzzy control.
 Design a fuzzy control membership function and rule base in accordance with the basic operation of the PV module.
 The membership function of the fuzzy control output consists of three types: increase (P: Positive), hold (ZE : zero) and decrease(N: negative)

The gain value of the PI controller is increased for quick response in transient conditions and reduced to reduce steady-state error in normal conditions. The VGPI proposed in this paper compares the tracking time at transient-state and the error in steady state with conventional MPPT methods for two scenarios (constant and changing conditions of solar radiation).
Under constant or varying conditions, the proportional gain (K_p) and integral gain (K_i) of the VGPI controller increased to the maximum value in the transient state and decreased to the minimum value in the steady state by the fuzzy control, and were continuously adjusted.
As a result, the VGPI controller proposed in this paper is about 14% better than the conventional PI and P&O in tracking speed, and the error in steady-state shown respectively 36.5% and 40% lower than PI and P&O. Even in conditions with varying solar radiation, the VGPI controller had excellent MPPT performance than other controllers because of continuous gain adjustment. VGPI controllers are able to adjust continuous gain values according to changing environments, and both transient and steady-state response performance was improved.
This method is expected to be applicable to various variable systems, as well as MPPT for solar power. Fuzzy control used in this paper requires continuous calculation according to changing environmental conditions. And the calculation of fuzzy control depends on the membership function and rule base, and the calculation amount can be greatly increased according to the environmental change conditions. This phenomenon degrades the MPPT performance.

Comments and Suggestions for Authors
Review Comments on sustainability-583644
Improvement of MPPT Control Performance Using Fuzzy Control and VGPI in the PV System for Microgrid

The abstract needs to be rephrased. Please take note of the sequence in terms of technical development for MPPT. First, PI controller is mentioned and its shortcomings are mentioned. In order to overcome the disadvantage of conventional PI controller, a new fuzzy PI controller is proposed in this paper.

-----
Reply-----

We revised the Abstract according to the reviewer's opinion.

CHANGE 1)
This paper proposes the method for maximum power point tracking(MPPT) of photovoltaic(PV) system. The conventional PI controller controls the system with fixed gains. Conventional PI controllers with fixed gains cannot satisfy both transient and steady-state. Therefore, to overcome the shortcomings of conventional PI controllers, this paper presents VGPI controllers that control the gain value of PI controllers using fuzzy control. Inputs of Fuzzy control used in the VGPI controller are the slope from the voltage-power characteristics of the PV module. This paper designs fuzzy control's membership functions and rule bases using the characteristics that the slope decrease in size as it approaches the maximum power point and increases as it gets farther. In addition, the gain of the PI controller is adjusted to increase in transient-state and decrease in steady-state in order to improve the error in steady-state and the tracking speed of maximum power point of the PV system. The performance of the VGPI controller has experimented in cases where the solar radiation is constant and the solar radiation varies, to compare with the performance of the P&O method, which is traditionally used most often in MPPT, and the performance of the PI controller, which is used most commonly in the industry field. Finally, the results from the experiment are presented and the results are analyzed.

On Page 1, delete the first 4 sentences from Line 29 to 33. Use only one sentence to mention application of solar energy and its development trend.

-----

Reply-----

We have removed this section according to the reviewer's comment.

CHANGE 1)
Fossil fuels have a problem of resource financing and environmental pollution, and interest in alternative energy sources is on the rise. In particular, greenhouse gases emitted by fossil fuels account for more than 80% of the total. Oil, natural gas, and coal are expected to be depleted due to the rising energy demand [1–2].

 Since the acronym of maximum power point tracking is given as MPPT at the beginning of this paper. MPPT should be applied through the text. It is not appropriate that maximum power point tracking is used sometime and MPPT is used some other times. Where maximum power point tracking appears, it must be replaced by MPPT through the paper.

-----

Reply-----

We modified the word according to the reviewer's opinion.

CHANGE 1)
maximum power point  MPP
maximum power point tracking  MPPT

 Line 48 on Page 2, remove ‘to control’. Line 85 on Page 2,

-----

Reply-----

We revised the content according to the reviewer's opinion.

CHANGE 1)
In addition, These methods use reference voltage[14-21], reference current[22-23], or duty ratio[24] for maximum power point tracking.

 it should be ‘ A solar cell consists of …’. Line 101 on Page 3, it should be ‘The diode current (Id) as shown in Figure 1 is given by the Shockley equation as follows: ‘. Line 104 on Page 3, what does [c] mean? After an equation, a stop ‘.’ must be added as a new sentence follows. In equation (1), G is not defined. Lines 115 ~ 116, “The relationship of … “ must be rephrased as the meaning is not clear.

-----

Reply-----

We deleted Chapter 2 in response to another reviewer's opinion that the contents of the paper are too long and easily accessible through other paper.

CHANGE 1)
2. Solar Cell Modeling
A solar cells consist of one ideal diode and a constant current source (I_ph). In reality, however, it is impossible to make an ideal diode; thus, a series resistor (R_s) and a parallel resistor (R_ph) representing the contact resistance and sheet resistance, respectively, of the surface layer must be considered. A part of the light incident on the surface of the solar cell is reflected from the surface, the light transmitted through the surface is absorbed in the solar cell, and the number of photons decreases exponentially.
Figure 1 shows the equivalent circuit of a solar cell.

Figure 1. Equivalent of solar cell

 The photocurrent (I_ph) is proportional to the solar radiation and is given by the following equation:
I_ph=(G/G_0 ) I_g0+J_0 (T_c-T_ref) (4)

Where G_o is the reference solar irradiance, G is the solar irradiance that enters the solar cell. I_g0 is the current at the reference solar irradiance (G_0), J_o is the temperature coefficient for photocurrent (I_ph), T_c is the temperature of the cell, and T_ref is the reference temperature of the cell. The reference radiation used in the equation representing photocurrent is mostly 1000[W/m^2]. The diode current (I_d) as shown in Figure 1 is given by the Shockley equation as follows.
I_d=I_0 [e^((q(v_pv+I_pv R_s )/nkT_c ) )-1] (2)
Here, V_pv+I_pv R_s represents the voltage of the diode, I_0 denotes the diode inverse saturation current, and q refers to the amount of electrons [q=1.602×〖10〗^(-19)  [C]]. V_pv and I_pv are the cell voltage and current, respectively, R_s is the series resistance, n is the ideal coefficient, and k is the Boltzmann constant (k=1.38×〖10〗^(-23)  [J/K]). Diode reverse saturation current I_0 is temperature-sensitive and can be expressed as:
I_0=I_d0 (T_c/T_ref )^3 e^[qE_g/nk (1/T_ref -1/T_c )]  (3)
Where I_d0 represents the diode inverse saturation current at the reference temperature, and T_ref  and T_c use the Kelvin temperature. Bandgap energy E_g of the silicon semiconductor constituting the solar cell can be expressed by the following equation:

E_g=1.17-(4.73×〖10〗^(-4)×T_c^2)/(T_c+636) (4)
The temperature of the solar cell (T_c) is proportional to the amount of solar radiation and can be expressed as:
T_c=273+T_a+((NOCT-20)/800)×G (5)
Where T_a represents the atmospheric temperature (℃) and NOCT (Nominal Operating Cell Temperature) denotes the nominal solar cell operating temperature. In the solar cell equivalent, circuit shown in Figure 1, the I_pv can be shown as follows.
I_pv=I_ph-I_o [e^(q(V_pv+I_pv×R_s )/(nkT_c ))-1]-(〖V_pv+I〗_pv R_s)/R_sh  (6)
Where R_sh represents the parallel resistance. The current in Equation (6) is common to the left and right equations, and the relationship of I_pv-V_pv can be expressed as [3, 35-38]:
f(I_pv,V_pv,G)=I_pv-{I_ph (G)-I_0 (G)[e^((q(V_pv+I_pv R_s )/(nkT_c (G) )) )-1]-(V_pv+(I_pv R_s)/R_sh )}=0 (7)

Flow charts shown in Figs. 6 and 8 should be associated with citation numbers if the charts are borrowed from the published articles.

-----

Reply-----

We have added a reference to the flowchart.

ADD 1)
Figure 5 shows the flow chart of Table 2[5-6, 10, 15].

ADD 2)
The flow chart of the IncCond method is shown in Figure 7 using the slope condition of the P-V curve and MPP variation according to the changing solar radiation in Figure 6[12, 14].

The conclusion section must be revised as the current writing is not a conclusion. It is rather written as an introduction. In conclusion, the authors need to give what have been achieved or obtained in this study, what the innovative approaches have been developed. A general overview or introduction to the domain knowledge must be removed. 

-----

Reply-----

We revised the conclusion according to the reviewer's opinion.

CHANGE 1)
This paper proposes a method for tracking the MPP of solar power generation. P&O and IncCond methods, which are commonly used MPPT methods, have limitations in performance improvement because they track a MPP by varying a voltage or a current with a constant magnitude.
In this paper, the VGPI controller is proposed to solve this problem. The characteristics of the VGPI controller presented in the paper are as follows.

 Based on the PI controller, which is most commonly used in industrial sites.
 The gain value of the PI controller adjusts according to the operating state using fuzzy control.
 Design a fuzzy control membership function and rule base in accordance with the basic operation of the PV module.
 The membership function of the fuzzy control output consists of three types: increase (P: Positive), hold (ZE : zero) and decrease(N: negative)

The gain value of the PI controller is increased for quick response in transient conditions and reduced to reduce steady-state error in normal conditions. The VGPI proposed in this paper compares the tracking time at transient-state and the error in steady state with conventional MPPT methods for two scenarios (constant and changing conditions of solar radiation).
Under constant or varying conditions, the proportional gain (K_p) and integral gain (K_i) of the VGPI controller increased to the maximum value in the transient state and decreased to the minimum value in the steady state by the fuzzy control, and were continuously adjusted.
As a result, the VGPI controller proposed in this paper is about 14% better than the conventional PI and P&O in tracking speed, and the error in steady-state shown respectively 36.5% and 40% lower than PI and P&O. Even in conditions with varying solar radiation, the VGPI controller had excellent MPPT performance than other controllers because of continuous gain adjustment. VGPI controllers are able to adjust continuous gain values according to changing environments, and both transient and steady-state response performance was improved.
This method is expected to be applicable to various variable systems, as well as MPPT for solar power. Fuzzy control used in this paper requires continuous calculation according to changing environmental conditions. And the calculation of fuzzy control depends on the membership function and rule base, and the calculation amount can be greatly increased according to the environmental change conditions. This phenomenon degrades the MPPT performance.

 It is not sure if the results presented are all correct. The authors must conduct a careful check again.

-----

Reply-----

The results presented in the paper are those tested by the experimental device in Figure 13. For clarity of the results of the experiment, the following are added:

ADD 1)
The PI controller used for comparison with VGPI uses 0.055 for proportional gain(K_p) and 0.007 for integral gain(K_i). Arduino's PWM ranges from 0 to 255, with 0 representing 0% and 255 representing 100% duty ratio. The P&O controller adjusts the PWM to a fixed size 3 to regulate the voltage at a constant rate.

ADD 2)
Figure 20(b) and 20(d) show the change in the gain value of the VGPI and PI controllers respectively. The gain value of the VGPI controller shows characteristic which gains are increasing in transient-state and decreasing until reaching a minimum value in steady-state.

ADD 3)

(b)

(d)
 (a) Power of VGPI MPPT method; (b) Gain of VGPI method; (c) Power of PI MPPT method; (d) Gain of PI method; (e) P&O MPPT method
Figure 20. Comparison of MPPT control response characteristics for solar radiation variation

Round 2

Reviewer 1 Report

Dear authors,

Thank you very much for your effort to improve the paper.

Best Regards

Author Response

Reply

First of all, we are very grateful that you’ve read our research work again and given us an appropriate comment. Our response is being highlighted in blue and we’ve attempted to make the paper more meaningful by supplementing "Experiment Result" section to increase the level of contribution. Thus, we respectfully request your re-review if possible. The changes or additions made are also being highlighted in Blue.

ADD)
Coelho, R. F.; Santos, W. M.; Martins, D. C. Influence of Power Converters on PV Maximum Power Point Tracking Efficiency. 2012 10Th IEEE/IAS International Conference on Industry Applications. 2012.

Coelho, R.F; Concer, F.M.; Martins, D. A. Study of the Basic DCDC converters Applied in Maximum Power Point Tracking. 10th Brazilian Power Electronics Conference, 2009, 673-679.

Ramki, T.; Tripathy, L. N. Comparison of Different DC-DC Converter for MPPT Application of Photovoltaic System. International Conference on Electrical, Electronics, Signals, Communication and Optimization (EESCO). Available online: https://www.academia.edu/24491040/Comparison_of_Different_DC-DC_Converter_for_MPPT_Application_of_Photovoltaic_System. (accessed on 26 September 2019).

Macaulay, J.; Zhou, Z. A Fuzzy Logical-Based Variable Step Size P&O MPPT Algorithm for Photovoltaic System. MDPI-energies. 2018, 11, 1340, 1-15.

Dolara, A.; Grimaccia, F.; Mussetta, M.; Ogliari, E.; Leva, S. An Evolutionary-Based MPPT Algorithm for Photovoltaic System under Dynamic Partial Shading. MDPI-applied science, 2018, 8, 558, 2-18.

Wan, Y.; Mao, M.; Zhou, L.; Zhang, Q.; Xi, X.; Zheng, C. A Novel Mature-Inspired Maximum Power Point Tracking(MPPT) Controller Based on SSA-GWO Algorithm for Partially Shaded Photovoltaic Systems. MDPI-electronics, 2019, 8, 680, 1-17.

Haque, A. Maximum Power Point Tracking(MPPT) Scheme for Solar Photovoltaic System. Energy Technology & Policy. 2014, 1, 115-122

Na, W.; Chen, P.; Kim, J.H.; An Improvement of a Fuzzy Logic-Controlled Maximum Power Point Tracking Algorithm for Photovoltaic Applications. MDPI-applied sciences. 2017. 7. 326, 1-17.

Li, C.; Chen, Y.; Zhou, D.; Liu, J.; Zeng, J. A High-Performance Adaptive Incremental Conductance MPPT Algorithm for Photovoltaic Systems. MDPI-energies. 2016, 9, 288, 1-17.

Piegari, L.; Rizzo, R.; Spina, I.; Tricoil, P. Optimized Adatpvie Perturb and Observe Maximum Power Point Tracking Control for Photovoltaic Generation. MDPI-energies, 2015. 8, 3418-3436.

Reviewer 2 Report

The paper has been improved based on the reviewers' comments, although the novelty of the paper is still very limited.

However, the design of the experiment is not correct. The output of the buck converter is a resistor load, and how the output voltage can be controlled when changing the input? The authors also need to measure the real PV maximum point point, and compare it with the result.

Author Response

(x) I don't feel qualified to judge about the English language and style

Reply-

The contents have been revised from the readers perspective with the assistance of a native English speaker and both the contribution and significance of the research are emphasized as well.

Comments and Suggestions for Authors
The paper has been improved based on the reviewers' comments, although the novelty of the paper is still very limited.

However, the design of the experiment is not correct. The output of the buck converter is a resistor load, and how the output voltage can be controlled when changing the input? The authors also need to measure the real PV maximum point point, and compare it with the result.

Reply-

First of all, we are very grateful that you’ve read our research work again and given us an appropriate comment. Our response is being highlighted in blue and we’ve attempted to make the paper more meaningful by supplementing "Experiment Result" section to increase the level of contribution. Thus, we respectfully request your re-review if possible. The changes or additions made are also being highlighted in Blue.

We added the following comments to Chapter 4, as well as related references, based on the comments of the review.

ADD 1)
The control performance of MPPT is verified by the speed at which maximum power is tracked and the magnitude of the error in steady state using voltage, current and power output from the PV module. In order to verify MPPT's performance in the paper, the experimental device was constructed with a PV module, Buck converter and resistive load. In the case of general solar power, it is connected by load to batteries or a DC bus of an inverter. If a resistive load is used to connect the load to the DC-DC converter, the effect of duty ratio control for MPPT control can be compared more accurately. Thus, a resistive load was used in this paper. In addition, the power consumed and the power generated by solar power is the same because the loads are constant when constructing resistive loads. Therefore, it is easy to get the power of PV system [52-57].
Experiments in solar power use solar simulators or use artificial light sources to construct a constant experimental environment. In the paper, a constant experimental environment was constructed using artificial lighting. Artificial light sources in the experimental environment can be used to maintain or change the test conditions. In addition, the same environmental conditions can be configured for different methods, so that the performance of the proposed method and the conventional method can be compared. The proposed method and conventional method compare the speed at which the maximum power point is tracked and the error at steady state. Since the environment is constructed using the same artificial light source, comparisons of output power, voltage and current can be a valid method for verifying peak power point tracking performance [58-61].

ADD 2)

(c) Artificial light

(a) Circuit diagram for experiments; (b) System for experiments; (c) Artificial light
Figure 13. Experimental setup for the MPPT control performance test of the PV system

Coelho, R. F.; Santos, W. M.; Martins, D. C. Influence of Power Converters on PV Maximum Power Point Tracking Efficiency. 2012 10Th IEEE/IAS International Conference on Industry Applications. 2012.

Coelho, R.F; Concer, F.M.; Martins, D. A. Study of the Basic DCDC converters Applied in Maximum Power Point Tracking. 10th Brazilian Power Electronics Conference, 2009, 673-679.

Ramki, T.; Tripathy, L. N. Comparison of Different DC-DC Converter for MPPT Application of Photovoltaic System. International Conference on Electrical, Electronics, Signals, Communication and Optimization (EESCO). Available online: https://www.academia.edu/24491040/Comparison_of_Different_DC-DC_Converter_for_MPPT_Application_of_Photovoltaic_System. (accessed on 26 September 2019).

Macaulay, J.; Zhou, Z. A Fuzzy Logical-Based Variable Step Size P&O MPPT Algorithm for Photovoltaic System. MDPI-energies. 2018, 11, 1340, 1-15.

Dolara, A.; Grimaccia, F.; Mussetta, M.; Ogliari, E.; Leva, S. An Evolutionary-Based MPPT Algorithm for Photovoltaic System under Dynamic Partial Shading. MDPI-applied science, 2018, 8, 558, 2-18.

Wan, Y.; Mao, M.; Zhou, L.; Zhang, Q.; Xi, X.; Zheng, C. A Novel Mature-Inspired Maximum Power Point Tracking(MPPT) Controller Based on SSA-GWO Algorithm for Partially Shaded Photovoltaic Systems. MDPI-electronics, 2019, 8, 680, 1-17.

Haque, A. Maximum Power Point Tracking(MPPT) Scheme for Solar Photovoltaic System. Energy Technology & Policy. 2014, 1, 115-122

Na, W.; Chen, P.; Kim, J.H.; An Improvement of a Fuzzy Logic-Controlled Maximum Power Point Tracking Algorithm for Photovoltaic Applications. MDPI-applied sciences. 2017. 7. 326, 1-17.

Li, C.; Chen, Y.; Zhou, D.; Liu, J.; Zeng, J. A High-Performance Adaptive Incremental Conductance MPPT Algorithm for Photovoltaic Systems. MDPI-energies. 2016, 9, 288, 1-17.

Piegari, L.; Rizzo, R.; Spina, I.; Tricoil, P. Optimized Adatpvie Perturb and Observe Maximum Power Point Tracking Control for Photovoltaic Generation. MDPI-energies, 2015. 8, 3418-3436.

Round 3

Reviewer 2 Report

Thanks for the revised paper, however, from the reviewer's point of view, the experimental setup won't work for PV systems. it is still not clear how the output voltage can be controlled when changing the input. 

Author Response

Reply-

First of all, we are very grateful that you’ve read our research work again and given us an appropriate comment. Our response is being highlighted in red and we’ve attempted to make the paper more meaningful by supplementing "Experiment Result" section to increase the level of contribution. The device for experimentation was newly constructed and experimented using it. And based on these results, we reconstructed the contents of Chapter 4. Thus, we respectfully request your re-review if possible. The changes or additions made are also being highlighted in red.

CHANGE 1)

Experiment Result

The control performance of MPPT is verified by the speed at which maximum power is tracked and the magnitude of the error in steady state using voltage, current and power output from the PV module. In order to verify MPPT's performance in the paper, the experimental device was constructed with a PV module, a Buck converter, a DC-DC step down converter and a battery.

Experiments in solar power use solar simulators or use artificial light sources to construct a constant experimental environment. In the paper, a constant experimental environment was constructed using artificial lighting. Artificial light sources in the experimental environment can be used to maintain or change the test conditions. In addition, the same environmental conditions can be configured for different methods, so that the performance of the proposed method and the conventional method can be compared. The proposed method and conventional method compare the speed at which the maximum power point is tracked and the error at steady state. Since the environment is constructed using the same artificial light source, comparisons of output power, voltage and current can be a valid method for verifying peak power point tracking performance [52-55].

Figure 13 shows the circuit diagram and control system for the MPPT control performance test of solar power generation. In this paper, MPPT control is controlled by the buck converter, and voltage and current are measured using the INA219 voltage current sensor. Switching of the buck converter was performed using P-channel MOSFET (F9530N) for high-side switching of the buck converter. P-channel MOSFET has a switching state of “on” when a "low" signal is inputted to the gate, so the NPN transistor (2N3904) and pull-up resistor (1kΩ) are used to control the buck converter.

DC-DC step down converter (KIS-3R33S) was used to maintain constant voltage for changes in the voltage of solar power, and the cell phone battery was charged.

(a)

(b)

(c)

(a) Circuit diagram for experiments; (b) System for experiments; (c) Artificial light

Figure 13. Experimental setup for the MPPT control performance test of the PV system

When MPPT control is performed using a buck converter, the voltage gradually decreases from the open-circuit voltage to the load voltage according to the PWM signal. In Figure 15, CH1 represents the PWM signal output from the controller, and CH2 denotes the voltage change of the PV module. The switching frequency of the controller used is 3.9 [kHz].

Figure 14. PWM signal and voltage PV module ( )

Figures 15 ~ 17 show the response characteristics of the VGPI, PI, and P&O methods under constant solar irradiation conditions.

The PI controller used for comparison with VGPI uses 0.035 for proportional gain( ) and 0.005 for integral gain( ). Arduino's PWM ranges from 0 to 255, with 0 representing 0% and 255 representing 100% duty ratio. The P&O controller adjusts the PWM to a fixed size 3 to regulate the voltage at a constant rate.

In Figure 15, (a) shows the voltage and current, (b) presents the output power, (c) illustrates the gain of the PI controller controlled by fuzzy control, (d) shows the control value ( ) and PWM signal for switching control of the DC-DC converter and (e) is output voltage controlled by step down converter. The gain of the PI controller in (C) is increased for fast tracking in transient state, and the gain value is decreased for improving accuracy and stability in steady state. The control value ( ) for tracking the MPP increases as the gain of the PI controller is adjusted according to the operating state and decreases in steady state. As a result, the variation of the PWM signal for switching of the DC-DC converter is reduced, and the power ripple is reduced; thus enabling more accurate MPPT. The output voltage in Figure (e) remains constant even as the voltage of solar power changes.

(a)

(b)

(c)

(d)

(e)

(a) Voltage ( ) and Current ( ) of the PV module; (b) Output Power of the PV module; (c) Proportional gain ( ) and Integral gain ( ) of the PI controller; (d) PWM signal and control value ( ) (e) Output Voltage

Figure 15. Response characteristics of the VGPI MPPT method

Figure 16 shows the response performance of the MPPT control of photovoltaic power generation using the PI controller. In particular, (c) shows the fixed gain of the PI controller. Although the PWM signal and the control value ( ) of Figure (d) are controlled according to operating state by PI control, the ripple of the output power increases because it is larger than the value of Figure 15 (d).

(a)

(b)

(c)

(d)

(e)

 (a) Voltage ( ) and Current ( ) of the PV module; (b) Output Power of the PV module; (c) Proportional gain ( ) and Integral gain ( ) of the PI controller; (d) PWM signal and control value ( ); (e) Output Voltage

Figure 16. Response characteristics of the PI MPPT method

Figure 17 shows the response characteristics of the most commonly used P&O method for MPPT control. In particular, (a) shows the voltage and current, (b) presents the output power, and (c) shows the PWM signal and control value ( ). Since the P&O method uses the fixed control value ( ) in both transient state and steady state, voltage in (a), power in (b), and PWM signal in (c) have a constant ripple magnitude.

(a)

(b)

(c)

(d)

(a) Voltage ( ) and Current ( ) of the PV module; (b) Output Power of the PV module; (c) PWM signal and control value ( ); (d) Output Voltage

Figure 17. Response characteristics of the P&O MPPT method

Figure 18 and Table 6 show a comparison of the power response characteristics in transient states of the VGPI, PI, and P&O methods in Figures 15 ~ 17. The time to trace the maximum power point in transient state was measured as the time to reach the average power (12.5[W]) of steady state, the results are shown in Table 6. As the results in Table 6 show, the VGPI controller has the most tracking speed with a high gain value in transient.

Figure 18. Comparison of response characteristics in transient state

Table 6. Comparison of rising time in Figure 18

VGPI

PI

P&O

Rising Time(sec)

5.94 (73.7%)

7.13 (88.4%)

8.06 (100%)

Figure 19 and Table 7 represent the magnified picture and characteristics of the steady-state portion of Figure 15-17. The VGPI controller had the lowest error because it had lower gain values in steady state, and the ripple was about 50% lower than the P&O method. Since the PI controller used a high gain value for fast tracking speed in transient conditions, steady-state error was rather higher than the P&O method.

(a)

(b)

(c)

(a) VGPI MPPT method; (b) PI MPPT method; (c) P&O MPPT method

Figure 19. Comparison of response characteristics in steady state

Table 7. Comparison of peak to peak in Figure 5

VGPI

PI

P&O

Min(W)

12.64

11.69

11.79

Max(W)

13.31

13.30

13.13

Peak to peak(W)

0.67(50.3%)

1.60(119.8%)

1.13(100%)

Figure 20 and Table 8 show MPPT control characteristics for changing conditions of solar radiation. Figure 20(a) represents the output of the VGPI, and Figure 20(b) represents the change in the gain value of the VGPI controller. The gain value of the VGPI controller represents a characteristic that is increasing in a transient state and is decreasing in steady state. Table 8 shows comparison of steady-state error for conditions with different solar irradiance. VGPI controllers show low steady-state error across all sections. Figure 20(d) represents the output voltage of the VGPI, PI, and P&O methods, with constant voltage output for the changing conditions of the solar radiation.

(a)

(b)

(c)

(d)

(e)

(a) Power of VGPI MPPT method; (b) Gain of VGPI method; (c) Power of PI MPPT method; (d) P&O MPPT method; (e) Output Voltage

Figure 20. Comparison of MPPT control response characteristics for solar radiation variation

Table 8. Comparison of steady-state error to solar radiation variation

VGPI

PI

P&O

11[sec] ~ 30[sec]

MIN [W]

3.95

3.96

3.46

MAX [W]

4.15

4.16

4.17

Peak to Peak [W]

0.20 (29.6%)

0.20 (29.6%)

0.71 (100%)

41[sec]~60[sec]

MIN [W]

12.47

10.82

11.78

MAX [W]

13.31

13.35

13.07

Peak to Peak [W]

0.84 (65.0%)

2.53 (195.9%)

1.29 (100%)
